# $L_2$-Nonexpansive Neural Networks

**Haifeng Qian & Mark N. Wegman**
IBM Research
Yorktown Heights, NY 10598, USA
`qianhaifeng,wegman@us.ibm.com`

## Abstract

This paper proposes a class of well-conditioned neural networks in which a unit amount of change in the inputs causes at most a unit amount of change in the outputs or any of the internal layers. We develop the known methodology of controlling Lipschitz constants to realize its full potential in maximizing robustness, with a new regularization scheme for linear layers, new ways to adapt nonlinearities and a new loss function. With MNIST and CIFAR-10 classifiers, we demonstrate a number of advantages. Without needing any adversarial training, the proposed classifiers exceed the state of the art in robustness against white-box $L_2$-bounded adversarial attacks. They generalize better than ordinary networks from noisy data with partially random labels. Their outputs are quantitatively meaningful and indicate levels of confidence and generalization, among other desirable properties.

## 1 Introduction

Artificial neural networks are often ill-conditioned systems in that a small change in the inputs can cause significant changes in the outputs (Szegedy et al., 2014). This results in poor robustness and vulnerability under adversarial attacks which has been reported on a variety of networks including image classification (Carlini & Wagner, 2017a; Goodfellow et al., 2014), speech recognition (Kreuk et al., 2018; Alzantot et al., 2018; Carlini & Wagner, 2018), image captioning (Chen et al., 2017) and natural language processing (Gao et al., 2018; Ebrahimi et al., 2017). These issues bring up both theoretical questions of how neural networks generalize (Kawaguchi et al., 2017; Xu & Mannor, 2012) and practical concerns of security in applications (Akhtar & Mian, 2018).

A number of remedies have been proposed for these issues and will be discussed in Section 4. White-box defense is particularly difficult and many proposals have failed. For example, Athalye et al. (2018) reported that out of eight recent defense works, only Madry et al. (2017) survived strong attacks. So far the mainstream and most successful remedy is that of adversarial training (Madry et al., 2017). However, as will be shown in Tables 1 and 2, the robustness by adversarial training diminishes when a white-box attacker (Carlini & Wagner, 2017a) is allowed to use more iterations.

This paper explores a different approach and demonstrates that a combination of the following three conditions results in enhanced robustness: 1) the Lipschitz constant of a network from inputs to logits is no greater than 1 with respect to the $L_2$-norm; 2) the loss function explicitly maximizes *confidence gap*, which is the difference between the largest and second largest logits of a classifier; 3) the network architecture restricts confidence gaps as little as possible. We will elaborate.

There are previous works that achieve the first condition (Cisse et al., 2017; Hein & Andriushchenko, 2017) or bound responses to input perturbations by other means (Kolter & Wong, 2017; Raghunathan et al., 2018; Haber & Ruthotto, 2017). For example, Parseval networks (Cisse et al., 2017) bound the Lipschitz constant by requiring each linear or convolution layer be composed of orthonormal filters. However, the reported robustness and guarantees are often under weak attacks or with low noise magnitude, and none of these works has demonstrated results that are comparable to adversarial training.

In contrast, we are able to build MNIST and CIFAR-10 classifiers, without needing any adversarial training, that exceed the state of the art (Madry et al., 2017) in robustness against white-box $L_2$-bounded adversarial attacks. The defense is even stronger if adversarial training is added. We will refer to these networks as $L_2$-*nonexpansive neural networks* (L2NNNs). Our advantage comes

from a set of new techniques: our weight regularization, which is key in enforcing the first condition, allows greater degrees of freedom in parameter training than the scheme in Cisse et al. (2017); a new loss function is specially designed for the second condition; we adapt various layers in new ways for the third condition, for example norm-pooling and two-sided ReLU, which will be presented later.

Let us begin with intuitions behind the second and third conditions. Consider a multi-class classifier. Let $g(\mathbf{x})$ denote its confidence gap for an input data point $\mathbf{x}$. If the classifier is a single L2NNN,[1] we have a guarantee[2] that the classifier will not change its answer as long as the input $\mathbf{x}$ is modified by no more than an $L_2$-norm of $g(\mathbf{x})/\sqrt{2}$. Therefore maximizing the average confidence gap directly boosts robustness and this motivates the second condition. To explain the third condition, let us introduce the notion of preserving distance: the distance between any pair of input vectors with two different labels ought to be preserved as much as possible at the outputs, while we do not care about the distance between a pair with the same label. Let $d(\mathbf{x_1}, \mathbf{x_2})$ denote the $L_2$-distance between the output logit-vectors for two input points $\mathbf{x_1}$ and $\mathbf{x_2}$ that have different labels and that are classified correctly. It is straightforward to verify the condition[3] of $g(\mathbf{x_1}) + g(\mathbf{x_2}) \leq \sqrt{2} \cdot d(\mathbf{x_1}, \mathbf{x_2})$. Therefore a network that maximizes confidence gaps well must be one that preserves distance well. Ultimately some distances are preserved while others are lost, and ideally the decision of which distance to lose is made by parameter training rather than by artifacts of network architecture. Hence the third condition involves distance-preserving architecture choices that leave the decision to parameter training as much as possible, and this motivates many of our design decisions such as Sections 2.2 and 2.3.

In practice we employ the strategy of divide and conquer and build each layer as a nonexpansive map with respect to the $L_2$-norm. It is straightforward to see that a feedforward network composed of nonexpansive layers must implement a nonexpansive map overall. How to adapt subtleties like recursion and splitting-reconvergence is included in the appendix.

Besides being robust against adversarial noises, L2NNNs have other desirable properties. They generalize better from noisy training labels than ordinary networks: for example, when 75% of MNIST training labels are randomized, an L2NNN still achieves 93.1% accuracy on the test set, in contrast to 75.2% from the best ordinary network. The problem of exploding gradients, which is common in training ordinary networks, is avoided because the gradient of any output with respect to any internal signal is bounded between -1 and 1. Unlike ordinary networks, the confidence gap of an L2NNN classifier is a quantitatively meaningful indication of confidence on individual data points, and the average gap is an indication of generalization.

## 2 $L_2$-NONEXPANSIVE NEURAL NETWORKS

This section describes how to adapt some individual operators in neural networks for L2NNNs. Discussions on splitting-reconvergence, recursion and normalization are in the appendix.

### 2.1 WEIGHTS

This section covers both the matrix-vector multiplication in a fully connected layer and the convolution calculation between input tensor and weight tensor in a convolution layer. The convolution calculation can be viewed as a set of vector-matrix multiplications: we make shifted copies of the input tensor and shuffle the copies into a set of small vectors such that each vector contains input entries in one tile; we reshape the weight tensor into a matrix by flattening all but the dimension of the output filters; then convolution is equivalent to multiplying each of the said small vectors with the flattened weight matrix. Therefore, in both cases, a basic operator is $\mathbf{y} = W\mathbf{x}$. To be a nonexpansive map with respect to the $L_2$-norm, a necessary and sufficient condition is

$$\mathbf{y}^{\mathrm{T}}\mathbf{y} \leq \mathbf{x}^{\mathrm{T}}\mathbf{x} \implies \mathbf{x}^{\mathrm{T}}W^{\mathrm{T}}W\mathbf{x} \leq \mathbf{x}^{\mathrm{T}}\mathbf{x}, \quad \forall \mathbf{x} \in \mathbb{R}^N$$
$$\rho\left(W^{\mathrm{T}}W\right) \leq 1 \tag{1}$$

where $\rho$ denotes the spectral radius of a matrix.

---

[1] This is only an example and we recommend building a classifier as multiple L2NNNs, see Section 2.4.

[2] See Lemma 1 in Appendix D for the proof of the guarantee.

[3] See Lemma 2 in Appendix D for the proof of the condition.

The exact condition of (1) is difficult to incorporate into training. Instead we use an upper bound:[4]

$$\rho\left(W^{\mathrm{T}}W\right) \leq b\left(W\right) \triangleq \min\left(r(W^{\mathrm{T}}W), r(WW^{\mathrm{T}})\right), \quad \text{where } r\left(M\right) = \max_i \sum_j |M_{i,j}| \quad (2)$$

The above is where our linear and convolution layers differ from those in Cisse et al. (2017): they require $WW^{\mathrm{T}}$ to be an identity matrix, and it is straightforward to see that their scheme is only one special case that makes $b\left(W\right)$ equal to 1. Instead of forcing filters to be orthogonal to each other, our bound of $b\left(W\right)$ provides parameter training with greater degrees of freedom.

One simple way to use (2) is replacing $W$ with $W' = W/\sqrt{b\left(W\right)}$ in weight multiplications, and this would enforce that the layer is strictly nonexpansive. Another method is described in the appendix.

As mentioned, convolution can be viewed as a first layer of making copies and a second layer of vector-matrix multiplications. With the above regularization, the multiplication layer is nonexpansive. Hence we only need to ensure that the copying layer is nonexpansive. For filter size of $K_1$ by $K_2$ and strides of $S_1$ by $S_2$, we simply divide the input tensor by a factor of $\sqrt{\lceil K_1/S_1 \rceil \cdot \lceil K_2/S_2 \rceil}$.

## 2.2 ReLU and others

Let us turn our attention to the third condition from Section 1. ReLU, tanh and sigmoid are nonexpansive but do not preserve distance well. This section presents a method that improves ReLU and is generalizable to other nonlinearities. A different approach to improve sigmoid is in the appendix.

To understand the weakness of ReLU, let us consider two input data points A and B, and suppose that a ReLU in the network receives two different negative values for A and B and outputs zero for both. Comparing the A-B distance before and after this ReLU layer, there is a distance loss and this particular ReLU contributes to it. We use *two-sided ReLU* which is a function from $\mathbb{R}$ to $\mathbb{R}^2$ and simply computes $\mathrm{ReLU}(x)$ and $\mathrm{ReLU}(-x)$. Two-sided ReLU has been studied in Shang et al. (2016) in convolution layers for accuracy improvement. It is straightforward to verify that two-sided ReLU is nonexpansive with respect to any $L_p$-norm and that it preserves distance in the above scenario. We will empirically verify its effectiveness in increasing confidence gaps in Section 3.

Two-sided ReLU is a special case of the following general technique. Let $f(x)$ be a nonexpansive and monotonically increasing scalar function, and note that ReLU, tanh and sigmoid all fit these conditions. We can define a function from $\mathbb{R}$ to $\mathbb{R}^2$ that computes $f(x)$ and $f(x) - x$. Such a new function is nonexpansive with respect to any $L_p$-norm[5] and preserves distance better than $f(x)$ alone.

## 2.3 Pooling

The popular max-pooling is nonexpansive, but does not preserve distance as much as possible. Consider a scenario where the inputs to pooling are activations that represent edge detection, and consider two images A and B such that A contains an edge that passes a particular pooling window while B does not. Inside this window, A has positive values while B has all zeroes. For this window, the A-B distance before pooling is the $L_2$-norm of A's values, yet if max-pooling is used, the A-B distance after pooling becomes the largest of A's values, which can be substantially smaller than the former. Thus we suffer a loss of distance between A and B while passing this pooling layer.

We replace max-pooling with norm-pooling, which was reported in Boureau et al. (2010) to occasionally increase accuracy. Instead of taking the max of values inside a pooling window, we take the $L_2$-norm of them. It is straightforward to verify that norm-pooling is nonexpansive[6] and would entirely preserve the $L_2$-distance between A and B in the hypothetical scenario above. Other $L_p$-norms can also be used. We will verify its effectiveness in increasing confidence gaps in Section 3.

If pooling windows overlap, we divide the input tensor by $\sqrt{K}$ where $K$ is the maximum number of pooling windows in which an entry can appear, similar to convolution layers discussed earlier.

---

[4]The spectral radius of a matrix is no greater than its natural $L_\infty$-norm. $W^{\mathrm{T}}W$ and $WW^{\mathrm{T}}$ have the same non-zero eigenvalues and hence the same spectral radius.

[5]See Lemma 4 in Appendix D for the proof of nonexpansiveness.

[6]See Lemma 5 in Appendix D for the proof of nonexpansiveness.

## 2.4 Loss function

For a classifier with $K$ labels, we recommend building it as $K$ overlapping L2NNNs, each of which outputs a single logit for one label. In an architecture with no split layers, this simply implies that these $K$ L2NNNs share all but the last linear layer and that the last linear layer is decomposed into $K$ single-output linear filters, one in each L2NNN. For a multi-L2NNN classifier, we have a guarantee[7] that the classifier will not change its answer as long as the input $\mathbf{x}$ is modified by no more than an $L_2$-norm of $g(\mathbf{x})/2$, where again $g(\mathbf{x})$ denotes the confidence gap. As mentioned in Section 1, a single-L2NNN classifier has a guarantee of $g(\mathbf{x})/\sqrt{2}$. Although this seems better on the surface, it is more difficult to achieve large confidence gaps. We will assume the multi-L2NNN approach.

We use a loss function with three terms, with trade-off hyperparameters $\gamma$ and $\omega$:

$$\mathcal{L} = \mathcal{L}_a + \gamma \cdot \mathcal{L}_b + \omega \cdot \mathcal{L}_c \tag{3}$$

Let $y_1, y_2, \cdots, y_K$ be outputs from the L2NNNs. The first loss term is

$$\mathcal{L}_a = \text{softmax-cross-entropy}\,(u_1 y_1, u_2 y_2, \cdots, u_K y_K, \text{label}) \tag{4}$$

where $u_1, u_2, \cdots, u_K$ are trainable parameters. The second loss term is

$$\mathcal{L}_b = \text{softmax-cross-entropy}\,(v y_1, v y_2, \cdots, v y_K, \text{label}) \tag{5}$$

where $v$ can be either a trainable parameter or a hyperparameter. Note that $u_1, u_2, \cdots, u_K$ and $v$ are not part of the classifier and are not used during inference. The third loss term is

$$\mathcal{L}_c = \frac{\text{average}\left(\log\left(1 - \text{softmax}\,(z y_1, z y_2, \cdots, z y_K)_{\text{label}}\right)\right)}{z} \tag{6}$$

where $z$ is a hyperparameter.

The rationale for the first loss term (4) is that it mimics cross-entropy loss of an ordinary network. If an ordinary network has been converted to L2NNNs by multiplying each layer with a small constant, its original outputs can be recovered by scaling up L2NNN outputs with certain constants, which is enabled by the formula (4). Hence this loss term is meant to guide the training process to discover any feature that an ordinary network can discover. The rationale for the second loss term (5) is that it is directly related to the classification accuracy. Multiplying L2NNN outputs uniformly with $v$ does not change the output label and only adapts to the value range of L2NNN outputs and drive towards better nominal accuracy. The third loss term (6) approximates average confidence gap: the log term is a soft measure of a confidence gap (for a correct prediction), and is asymptotically linear for larger gap values. The hyperparameter $z$ controls the degree of softness, and has relatively low impact on the magnitude of loss due to the division by $z$; if we increase $z$ then (6) asymptotically becomes the average of minus confidence gaps for correct predictions and zeroes for incorrect predictions. Therefore loss (6) encourages large confidence gaps and yet is smooth and differentiable.

A notable variation of (3) is one that combines with adversarial training. Our implementation applies the technique of Madry et al. (2017) on the first loss term (4): we use distorted inputs in calculating $\mathcal{L}_a$. The results are reported in Tables 1 and 2 as Model 4. Another possibility is to use distorted inputs in calculating $\mathcal{L}_a$ and $\mathcal{L}_b$, while $\mathcal{L}_c$ should be based on original inputs.

## 3 Experiments

Experiments are divided into three groups to study different properties of L2NNNs. Our MNIST and CIFAR-10 classifiers are available at
http://researcher.watson.ibm.com/group/9298

### 3.1 Robustness

This section evaluates robustness of L2NNN classifiers for MNIST and CIFAR-10 and compares against the state of the art Madry et al. (2017). The robustness metric is accuracy under white-box non-targeted $L_2$-bounded attacks. The attack code of Carlini & Wagner (2017a) is used. We

---

[7]See Lemma 6 in Appendix D for the proof of the guarantee. The guarantee in either Lemma 1 or Lemma 6 is only a loose guarantee and it has been shown in Hein & Andriushchenko (2017) that a larger guarantee exists by analyzing local Lipschitz constants, though it is expensive to compute.

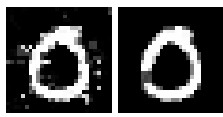

Figure 1: Attacks on Model 2 found after 1K and 10K iterations: the same 0 recognized as 5.

downloaded the classifiers[8] of Madry et al. (2017) and report their robustness against $L_2$-bounded attacks in Tables 1 and 2.[9] Note that their defense diminishes as the attacks are allowed more iterations. Figure 1 illustrates one example of this effect: the first image is an attack on MNIST Model 2 (0 recognized as 5) found after 1K iterations, with noise $L_2$-norm of 4.4, while the second picture is one found after 10K iterations, the same 0 recognized as 5, with noise $L_2$-norm of 2.1. We hypothesize that adversarial training alone provides little absolute defense at the noise levels used in the two tables: adversarial examples still exist and are only more difficult to find. The fact that in Table 2 Model 2 accuracy is lower in the 1000x10 row than the 10K row further supports our hypothesis.

Table 1: Accuracies of MNIST classifiers under white-box non-targeted attacks with noise $L_2$-norm limit of 3. MaxIter is the max number of iterations the attacker uses. Model 1 is an ordinarily trained model. Model 2 is the model from Madry et al. (2017). Model 3 is L2NNN without adversarial training. Model 4 is L2NNN with adversarial training.

| MaxIter | Model1 | Model2 | Model3 | Model4 |
|---|---|---|---|---|
| Natural | 99.1% | 98.5% | 98.7% | 98.2% |
| 100 | 70.2% | 91.7% | 77.6% | 75.6% |
| 1000 | 0.05% | 51.5% | 20.3% | 24.4% |
| 10K | 0% | 16.0% | 20.1% | 24.4% |
| 100K | 0% | 9.8% | 20.1% | 24.4% |
| 1M | 0% | 7.6% | 20.1% | 24.4% |

Table 2: Accuracies of CIFAR-10 classifiers under white-box non-targeted attacks with noise $L_2$-norm limit of 1.5. MaxIter is the max number of iterations the attacker uses, and 1000x10 indicates 10 runs each with 1000 iterations. Model 1 is an ordinarily network. Model 2 is the model from Madry et al. (2017). Model 3 is L2NNN without adversarial training. Model 4 is L2NNN with adversarial training.

| MaxIter | Model1 | Model2 | Model3 | Model4 |
|---|---|---|---|---|
| Natural | 95.0% | 87.1% | 79.2% | 77.2% |
| 100 | 0% | 13.9% | 10.2% | 20.8% |
| 1000 | 0% | 9.4% | 10.1% | 20.4% |
| 10K | 0% | 9.0% | 10.1% | 20.4% |
| 1000x10 | 0% | 8.7% | 10.1% | 20.4% |
| 100K | 0% | NA | 10.1% | 20.4% |

In contrast, the defense of the L2NNN models remain constant when the attacks are allowed more iterations, specifically MNIST Models beyond 10K iterations and CIFAR-10 Models beyond 1000

---

[8]At `github.com/MadryLab/mnist_challenge` and `github.com/MadryLab/cifar10_challenge`. These models (Model 2's in Tables 1 and 2) were built by adversarial training with $L_\infty$-bounded adversaries (Madry et al., 2017). To the best of our knowledge, Tsipras et al. (2019) from the same lab is the only paper in the literature that reports on models trained with $L_2$-bounded adversaries, and it reports that training with $L_2$-bounded adversaries resulted in weaker $L_2$ robustness than the $L_2$ robustness results from training with $L_\infty$-bounded adversaries in Madry et al. (2017). Therefore we choose to compare against the best available models, even though they were trained with $L_\infty$-bounded adversaries. Note also that our own Model 4's in Tables 1 and 2 are trained with the same $L_\infty$-bounded adversaries.

[9]In reading Tables 1 and 2, it is worth remembering that the norm of after-attack accuracy is zero, and for example the 7.6% on MNIST is currently the state of the art.

iterations. The reason is that L2NNN classifiers achieve their defense by creating a confidence gap between the largest logit and the rest, and that half of this gap is a lower bound of $L_2$-norm of distortion to the input data in order to change the classification. Hence L2NNN's defense comes from a minimum-distortion guarantee. Although adversarial training alone may also increase the minimum distortion limit for misclassification, as suggested in Carlini et al. (2017) for a small network, that limit likely does not reach the levels used in Tables 1 and 2 and hence the defense depends on how likely the attacker can reach a lower-distortion misclassification. Consequently when the attacks are allowed to make more attempts the defense with guarantee stands while the other diminishes.

For both MNIST and CIFAR-10, adding adversarial training boosts the robustness of Model 4. We hypothesize that adversarial training lowers local Lipschitz constants in certain parts of the input space, specifically around the training images, and therefore makes local robustness guarantees larger (Hein & Andriushchenko, 2017). To test this hypothesis on MNIST Models 3 and 4, we measure the average $L_2$-norm of their Jacobian matrices, averaged over the first 1000 images in the test set, and the results are 1.05 for Model 3 and 0.83 for Model 4. Note that the $L_2$-norm of Jacobian can be greater than 1 for multi-L2NNN classifiers. These measurements are consistent with, albeit does not prove, the hypothesis.

Table 3: Ablation studies: MNIST model without weight regularization; one without $\mathcal{L}_c$ loss; one with max-pooling instead of norm-pooling; one without two-sided ReLU; Gap is average confidence gap. R-Accu is under attacks with 1000 iterations and with noise $L_2$-norm limit of 3.

|  | Accu. | Gap | R-Accu. |
|---|---|---|---|
| no weight reg. | 99.4% | 68.3 | 0% |
| no $\mathcal{L}_c$ loss | 99.2% | 2.2 | 8.9% |
| no norm-pooling | 98.8% | 1.3 | 9.9% |
| no two-sided ReLU | 98.0% | 2.5 | 15.1% |

To test the effects of various components of our method, we build models for each of which we disable a different technique during training. The results are reported in Table 3. To put the confidence gap values in context, our MNIST Model 3 has an average gap of 2.8. The first one is without weight regularization of Section 2.1 and it becomes an ordinary network which has little defense against adversarial attacks; its large average confidence gap is meaningless. For the second one we remove the third loss term (6) and for the third one we replace norm-pooling with regular max-pooling, both resulting in smaller average confidence gap and less defense against attacks. For the fourth one, we replace two-sided ReLU with regular ReLU, and this leads to degradation in nominal accuracy, average confidence gap and robustness. Parseval networks (Cisse et al., 2017) can be viewed as models without $\mathcal{L}_c$ term, norm-pooling or two-sided ReLU, and with a more restrictive scheme for weight matrix regularization.

Model 3 in Table 1 and the second row of Table 3 are two points along a trade-off curve that are controllable by varying hyperparameter $\omega$ in loss function (3). Other trade-off points have nominal accuracy and under-attack accuracy of (98.8%,19.1%), (98.4%,22.6%) and (97.9%,24.7%) respectively. Similar trade-offs have been reported by other robustness works including adversarial training (Tsipras et al., 2019) and adversarial polytope (Wong et al., 2018). It remains an open question whether such trade-off is a necessary part of life, and please see Section 3.3 for further discussion on the L2NNN trade-off.

Table 4: Accuracy of L2NNN classifiers under white-box non-targeted attacks with 1000 iterations and with noise $L_\infty$-norm limit of $\epsilon$.

|  | $\epsilon$ | Model3 | Model4 |
|---|---|---|---|
| MNIST | 0.1 | 90.9% | 92.4% |
| MNIST | 0.3 | 7.0% | 44.0% |
| CIFAR-10 | $^8/_{256}$ | 32.3% | 42.5% |

Although we primarily focus on defending against $L_2$-bounded adversarial attacks in this work, we achieve some level of robustness against $L_\infty$-bounded attacks as a by-product. Table 4 shows

our results, again measured with the attack code of Carlini & Wagner (2017a). The $\epsilon$ values match those used in Raghunathan et al. (2018); Kolter & Wong (2017); Madry et al. (2017). Our MNIST $L_\infty$ results are on par with Raghunathan et al. (2018); Kolter & Wong (2017) but not as good as Madry et al. (2017). Our CIFAR-10 Model 4 is on par with Madry et al. (2017) for $L_\infty$ defense.

## 3.2 MEANINGFUL OUTPUTS

This section discusses how to understand and utilize L2NNNs' output values. We observe strong correlation between the confidence gap of L2NNN and the magnitude of distortion needed to force it to misclassify, and images are included in appendix.

In the next experiment, we sort test data by the confidence gap of a classifier on each image. Then we divide the sorted data into 10 bins and report accuracy separately on each bin in Figure 2. We repeat this experiment for Model 2 (Madry et al., 2017) and our Model 3 of Tables 1 and 2. Note that the L2NNN model shows better correlation between confidence and robustness: for MNIST our first bin is 95% robust and second bin is 67% robust. This indicates that the L2NNN outputs are much more quantitatively meaningful than those of ordinary neural networks.

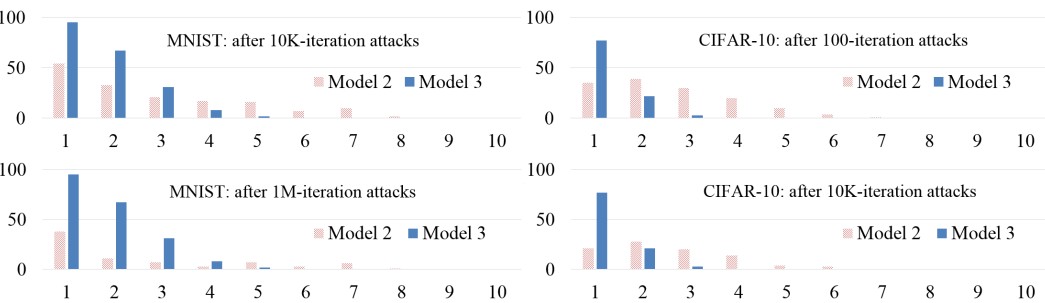

Figure 2: Accuracy percentages of classifiers on test data bin-sorted by the confidence gap.

It is an important property that an L2NNN has an easily accessible measurement on how robust its decisions are. Since robustness is easily measurable, it can be optimized directly, and we believe that this is the primary reason that we can demonstrate the robustness results of Tables 1 and 2. This can also be valuable in real-life applications where we need to quantify how reliable a decision is.

One of the other practical implications of this property is that we can form hybrid models which use L2NNN outputs when the confidence is high and a different model when the confidence of the L2NNN is low. This creates another dimension of trade-off between nominal accuracy and robustness that one can take advantage of in an application. We built such a hybrid model for MNIST with the switch threshold of 1.0 and achieved nominal accuracy of 99.3%, where only 6.9% of images were delegated to the alternative classifier. We built such a hybrid model for CIFAR-10 with the switch threshold of 0.1 and achieved nominal accuracy of 89.4%, where 25% of images were delegated. To put these threshold values in context, MNIST Model 3 has an average gap of 2.8 and CIFAR-10 Model 3 has an average gap of 0.34. In other words, if for a data point the L2NNN confidence gap is substantially below average, the classification is delegated to the alternative classifier, and this way we can recover nominal accuracy at a moderate cost of robustness.

## 3.3 GENERALIZATION VERSUS MEMORIZATION

This section studies L2NNN's generalization through a noisy-data experiment where we randomize some or all MNIST training labels. The setup is similar to Zhang et al. (2017), except that we added three scenarios where 25%, 50% and 75% of training labels are scrambled.

Table 5 shows the comparison between L2NNNs and ordinary networks. Dropout rate and weight-decay weight are tuned for each WD/DR run, and each WD+DR+ES run uses the combined hyper-parameters from its row. In early-stopping runs, 5000 training images are withheld as validation set and training stops when loss on validation set stops decreasing. The L2NNNs do not use weight decay, dropout or early stopping. L2NNNs achieve the best accuracy in all three partially-scrambled

Table 5: Accuracy comparison of MNIST classifiers that are trained on noisy data. Rand is the percentage of training labels that are randomized. WD is weight decay. DR is dropout. ES is early stopping. Gap1 is L2NNN's average confidence gap on training set and Gap2 is that on test set.

| Rand | Ordinary network | | | | | L2NNN | | |
|------|---------|------|------|------|----------|------|------|------|
| | Vanilla | WD | DR | ES | WD+DR+ES | | Gap1 | Gap2 |
| 0 | 99.4% | 99.0% | 99.2% | 99.0% | 99.3% | 98.7% | 2.84 | 2.82 |
| 25% | 90.4% | 91.1% | 91.8% | 96.2% | 98.0% | 98.5% | 0.64 | 0.63 |
| 50% | 65.5% | 67.7% | 72.6% | 81.0% | 88.3% | 96.0% | 0.58 | 0.60 |
| 75% | 41.5% | 44.9% | 41.8% | 75.2% | 66.4% | 93.1% | 0.86 | 0.89 |
| 100% | 9.7% | 9.1% | 9.4% | NA | NA | 11.9% | 0.09 | 0.01 |

Table 6: Training-accuracy-versus-confidence-gap trade-off points of L2NNNs on 50%-scrambled MNIST training labels.

| on training set | | on test set | |
|------|------|------|------|
| Accu. | Gap | Accu. | Gap |
| 98.7% | 0.17 | 79.0% | 0.12 |
| 96.5% | 0.21 | 79.3% | 0.18 |
| 89.4% | 0.22 | 86.3% | 0.20 |
| 70.1% | 0.36 | 93.4% | 0.37 |
| 66.1% | 0.45 | 93.7% | 0.47 |
| 59.8% | 0.58 | 96.0% | 0.60 |

scenarios, and it is remarkable that an L2NNN can deliver 93.1% accuracy on test set when three quarters of training labels are random. More detailed data and discussions are in the appendix.

To illustrate why L2NNNs generalize better than ordinary networks from noisy data, we show in Table 6 trade-off points between accuracy and confidence gap on the 50%-scrambled training set. These trade-off points are achieved by changing hyperparameters $\omega$ in (3) and $v$ in (5). In a noisy training set, there exist data points that are close to each other yet have different labels. For a pair of such points, if an L2NNN is to classify both points correctly, the two confidence gaps must be small. Therefore, in order to achieve large average confidence gap, an L2NNN must misclassify some of the training data. In Table 6, as we adjust the loss function to favor larger average gap, the L2NNNs are forced to make more and more mistakes on the training set. The results suggest that loss is minimized when an L2NNN misclassifies some of the scrambled labels while fitting the 50% original labels with large gaps, and parameter training discovers this trade-off automatically. Hence we see in Table 6 increasing accuracies and gaps on the test set. The above is a trade-off between memorization (training-set accuracy) and generalization (training-set average gap), and we hypothesize that L2NNN's trade-off between nominal accuracy and robustness, reported in Section 3.1, is due to the same mechanism. To be fair, dropout and early stopping are also able to sacrifice accuracy on a noisy training set, however they do so through different mechanisms that tend to be brittle, and Table 5 suggests that L2NNN's mechanism is superior. More discussions and the trade-off tables for 25% and 75% scenarios are in the appendix.

Another interesting observation is that the average confidence gap dramatically shrinks in the last row of Table 5 where the training is pure memorization. This is not surprising again due to training data points that are close to each other yet have different labels. The practical implication is that after an L2NNN model is trained, one can simply measure its average confidence gap to know whether and how much it has learned to generalize rather than to memorize the training data.

## 4 RELATED WORK

Adversarial defense is a well-known difficult problem (Szegedy et al., 2014; Goodfellow et al., 2014; Carlini & Wagner, 2017a; Athalye et al., 2018; Gilmer et al., 2018). There are many avenues

to defense (Carlini & Wagner, 2017b; Meng & Chen, 2017), and here we will focus on defense works that fortify a neural network itself instead of introducing additional components.

The mainstream approach has been adversarial training, where examples of successful attacks on a classifier itself are used in training (Tramèr et al., 2017; Zantedeschi et al., 2017). The work of Madry et al. (2017) has the best results to date and effectively flattens gradients around training data points, and, prior to our work, it is the only work that achieves sizable white-box defense. It has been reported in Carlini et al. (2017) that, for a small network, adversarial training indeed increases the average minimum $L_1$-norm and $L_\infty$-norm of noise needed to change its classification. However, in view of results of Tables 1 and 2, adversarial-training results may be susceptible to strong attacks. The works of Drucker & Le Cun (1992); Ross & Doshi-Velez (2017) are similar to adversarial training in aiming to flatten gradients around training data set but use different mechanisms.

While the above approaches fortify a network around training data points, others aim to bound a network's responses to input perturbations over the entire input space. For example, Haber & Ruthotto (2017) models ResNet as an ordinary differential equation and derive stability conditions. Other examples include Kolter & Wong (2017); Raghunathan et al. (2018); Wong et al. (2018) which achieved provable guarantees against $L_\infty$-bounded attacks. However there exist scalability issues with respect to network depth, and the reported results so far are against relatively weak attacks or low noise magnitude. As shown in Table 4, we can match their measured $L_\infty$-bounded defense.

Controlling Lipschitz constants also regularizes a network over the entire input space. Szegedy et al. (2014) is the seminal work that brings attention to this topic. Bartlett et al. (2017) proposes the notion of spectrally-normalized margins as an indicator of generalization, which are strongly related to our confidence gap. Pascanu et al. (2013) studies the role of the spectral radius of weight matrices in the vanishing and the exploding gradient problems. Yoshida & Miyato (2017) proposes a method to regularize the spectral radius of weight matrices and shows its effect in reducing generalization gap. The work on Parseval networks (Cisse et al., 2017) shows that it is possible to control Lipschitz constants of neural networks through regularization. The core of their work is to constrain linear and convolution layer weights to be composed of Parseval tight frames, i.e., orthonormal filters, and thereby force the Lipschitz constant of these layers to be 1; they also propose to restrict aggregation operations. The reported robustness results of Cisse et al. (2017), however, are much weaker than those by adversarial training in Madry et al. (2017). We differ from Parseval networks in a number of ways. Our linear and convolution layers do not require filters to be orthogonal to each other and subsume Parseval layers as a special case, and therefore provide more freedom to parameter training. We use non-standard techniques, e.g. two-sided ReLU, to modify various network components to maximize confidence gaps while keeping the network nonexpansive, and we propose a new loss function for the same purpose. We are unable to obtain Parseval networks for a direct comparison, however it is possible to get a rough idea of what the comparison might be by looking at Table 3 which shows the impacts of those new techniques. The work of Hein & Andriushchenko (2017) makes an important point regarding guarantees provided by local Lipschitz constants, which helps explain many observations in our results, including why adversarial training on L2NNNs leads to lasting robustness gains. The regularization proposed by Hein & Andriushchenko (2017) however is less practical and again introduces reliance on the coverage of training data points.

## 5 CONCLUSIONS AND FUTURE WORK

In this work we have presented $L_2$-nonexpansive neural networks which are well-conditioned systems by construction. Practical techniques are developed for building these networks. Their properties are studied through experiments and benefits demonstrated, including that our MNIST and CIFAR-10 classifiers exceed the state of the art in robustness against white-box adversarial attacks, that they are robust against partially random training labels, and that they output confidence gaps which are strongly correlated with robustness and generalization. There are a number of future directions, for example, other applications of L2NNN, L2NNN-friendly neural network architectures, and the relation between L2NNNs and interpretability.

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

# A  $L_2$-NONEXPANSIVE NETWORK COMPONENTS

## A.1  ADDITIONAL METHODS FOR WEIGHT REGULARIZATION

There are numerous ways to utilize the bound of (2). The main text describes a simple method of using $W' = W/\sqrt{b\,(W)}$ to enforce strict nonexpansiveness. The following is an alternative.

Approximate nonexpansiveness can be achieved by adding a penalty to the loss function whenever $b\,(W)$ exceeds 1, for example:

$$\mathcal{L}_W = \min\left(l(W^{\mathrm{T}}W), l(WW^{\mathrm{T}})\right), \text{ where } l\,(M) = \sum_i \max\left(\sum_j |M_{i,j}| - 1, 0\right) \qquad (7)$$

The sum of (7) losses over all layers becomes a fourth term in the loss function (3), multiplied with one additional hyperparameter. This would lead to an approximate L2NN with trade-offs between how much its layers violate (1) with surrogate (2) versus other objectives in the loss function.

In practice, we have found that it is beneficial to begin neural network training with the regularization scheme of (7), which allows larger learning rates, and switch to the first scheme of using $W'$, which avoids artifacts of an extra hyperparameter, when close to convergence. Of course if the goal is building approximate L2NNNs one can use (7) all the way.

## A.2  SIGMOID AND OTHERS

Sigmoid is nonexpansive as is, but does not preserve distance as much as possible. A better way is to replace sigmoid with the following operator

$$s\,(x) = t \cdot \mathrm{sigmoid}\left(\frac{4x}{t}\right) \qquad (8)$$

where $t > 0$ is a trainable parameter and each neuron has its own $t$. In general, the requirement for any scalar nonlinearity is that its derivative is bounded between -1 and 1. If a nonlinearity violates this condition, a shrinking multiplier can be applied. If the actual range of derivative is narrower, as in the case of sigmoid, an enlarging multiplier can be applied to preserve distance.

For further improvement, (8) can be combined with the general form of the two-sided ReLU of Section 2.2. Then the new nonlinearity is a function from $\mathbb{R}$ to $\mathbb{R}^2$ that computes $s(x)$ and $s(x) - x$.

## A.3  SPLITTING AND RECONVERGENCE

There are different kinds of splitting in neural networks. Some splitting is not followed by reconvergence. For example, a classifier may have common layers followed by split layers for each label, and such an architecture can be viewed as multiple L2NNNs that overlap at the common layers and each contain one stack of split layers. In such cases, no modification is needed because there is no splitting within each individual L2NNN.

Some splitting, however, is followed by reconvergence. In fact, convolution and pooling layers discussed earlier can be viewed as splitting, and reconvergence happens at the next layer. Another common example is skip-level connections such as in ResNet. Such splitting should be viewed as making two copies of a certain vector. Let the before-split vector be $\mathbf{x}_0$, and we make two copies as

$$\begin{aligned} \mathbf{x}_1 &= t \cdot \mathbf{x}_0 \\ \mathbf{x}_2 &= \sqrt{1 - t^2} \cdot \mathbf{x}_0 \end{aligned} \qquad (9)$$

where $t \in [0, 1]$ is a trainable parameter.

In the case of ResNet, the reconvergence is an add operator, which should be treated as vector-matrix multiplication as in Section 2.1, but with much simplified forms. Let $\mathbf{x}_1$ be the skip-level connections and $f\,(\mathbf{x}_2)$ be the channels of convolution outputs to be added with $\mathbf{x}_1$, we perform the addition as

$$\mathbf{y} = t \cdot \mathbf{x}_1 + \sqrt{1 - t^2} \cdot f\,(\mathbf{x}_2) \qquad (10)$$

where $t \in [0, 1]$ is a trainable parameter and could be a common parameter with (9).

ResNet-like reconvergence is referred to as aggregation layers in Cisse et al. (2017) and a different formula was used:

$$\mathbf{y} = \alpha \cdot \mathbf{x}_1 + (1 - \alpha) \cdot f(\mathbf{x}_2) \tag{11}$$

where $\alpha \in [0, 1]$ is a trainable parameter. Because splitting is not modified in Cisse et al. (2017), their scheme may seem approximately equivalent to ours if a common $t$ parameter is used for (9) and (10). However, there is a substantial difference: in many ResNet blocks, $f(\mathbf{x}_2)$ is a subset of rather than all of the output channels of convolution layers, and our scheme does not apply the shrinking factor of $\sqrt{1 - t^2}$ on channels that are not part of $f(\mathbf{x}_2)$ and therefore better preserve distances. In contrast, because splitting is not modified, at reconvergence the scheme of Cisse et al. (2017) must apply the shrinking factor of $1 - \alpha$ on all outputs of convolution layers, regardless of whether a channel is part of the aggregation or not. To state the difference in more general terms, our scheme enables splitting and reconvergence at arbitrary levels of granularity and multiplies shrinking factors to only the necessary components. We can also have a different $t$ per channel or even per entry.

To be fair, the scheme of Cisse et al. (2017) has an advantage of being nonexpansive with respect to any $L_p$-norm. However, for $L_2$-norm, it is inferior to ours in preserving distances and maximizing confidence gaps.

### A.4 RECURSION

There are multiple ways to interpret recurrent neural networks (RNN) as L2NNNs. One way is to view an unrolled RNN as multiple overlapping L2NNNs where each L2NNN generates the output at one time step. Under this interpretation, nothing special is needed and recurrent inputs to a neuron are simply treated as ordinary inputs.

Another way to interpret an RNN is to view unrolled RNN as a single L2NNN that generates outputs at all time steps. Under this interpretation, recurrent connections are treated as splitting at their sources and should be handled as in (9).

### A.5 NORMALIZATION

Normalization operations are limited in an L2NNN. Subtracting mean is nonexpansive and allowed, and subtract-mean operation can be performed on arbitrary subsets of any layer. Subtracting batch mean is also allowed because it can be viewed as subtracting a bias parameter. However, scaling, e.g., division by standard deviation or batch standard deviation is only allowed if the multiplying factors are between -1 and 1. To satisfy this in practice, one simple method is to divide all multiplying factors in a normalization layer by the largest of their absolute values.

## B MNIST IMAGES

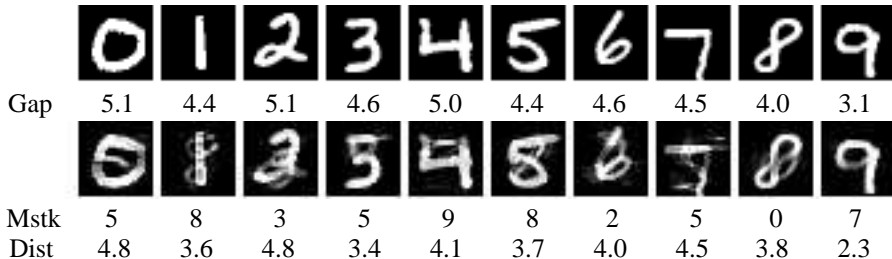

| | | | | | | | | | |
|---|---|---|---|---|---|---|---|---|---|
| Gap | 5.1 | 4.4 | 5.1 | 4.6 | 5.0 | 4.4 | 4.6 | 4.5 | 4.0 | 3.1 |

| | | | | | | | | | |
|---|---|---|---|---|---|---|---|---|---|
| Mstk | 5 | 8 | 3 | 5 | 9 | 8 | 2 | 5 | 0 | 7 |
| Dist | 4.8 | 3.6 | 4.8 | 3.4 | 4.1 | 3.7 | 4.0 | 4.5 | 3.8 | 2.3 |

Figure 3: Original and distorted images of MNIST digits in test set with the largest confidence gaps. Mstk denotes the misclassified labels. Dist denotes the $L_2$-norm of the distortion noise.

Let us begin by showing MNIST images with the largest confidence gaps in Figure 3 and those with the smallest confidence gaps in Figure 4. They include images before and after attacks as well as Model 3's confidence gap, the misclassified label and $L_2$-norm of the added noise. The images with large confidence gaps seem to be ones that are most different from other digits, while some

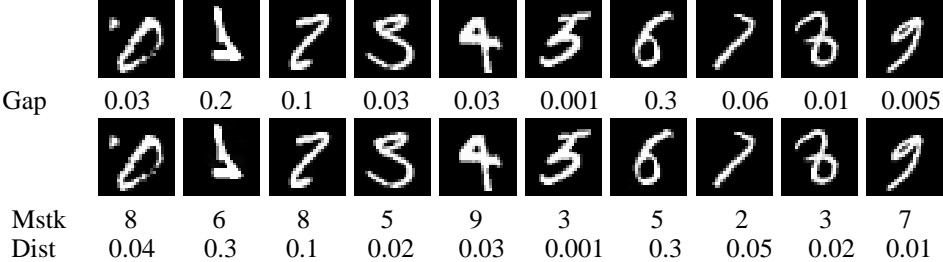

| | | | | | | | | | | |
|---|---|---|---|---|---|---|---|---|---|---|
| Gap | 0.03 | 0.2 | 0.1 | 0.03 | 0.03 | 0.001 | 0.3 | 0.06 | 0.01 | 0.005 |

| | | | | | | | | | | |
|---|---|---|---|---|---|---|---|---|---|---|
| Mstk | 8 | 6 | 8 | 5 | 9 | 3 | 5 | 2 | 3 | 7 |
| Dist | 0.04 | 0.3 | 0.1 | 0.02 | 0.03 | 0.001 | 0.3 | 0.05 | 0.02 | 0.01 |

Figure 4: Original and distorted images of MNIST digits in test set with the smallest confidence gaps. Mstk denotes the misclassified output label. Dist denotes the $L_2$-norm of the distortion noise.

of the images with small confidence gaps are genuinely ambiguous. It's worth noting the strong correlation between the confidence gap of L2NNN and the magnitude of distortion needed to force it to misclassify. Also note that our guarantee states that the minimum $L_2$-norm of noise is half of the confidence gap, but in reality the needed noise is much stronger than the guarantee. The reason is that the true local guarantee is in fact larger due to local Lipschitz constants, as pointed out by Hein & Andriushchenko (2017).

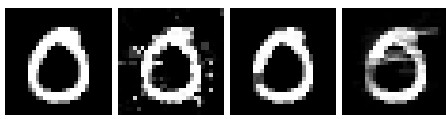

Figure 5: Original image of 0; attack on Model 2 (Madry et al., 2017) found after 1K iterations; attack on Model 2 found after 10K iterations; attack on Model 3 (L2NNN) found after 1M iterations. The latter three all lead to misclassification as 5.

Figure 5 shows additional details regarding the example in Figure 1. The first image is the original image of a zero. The second image is an attack on Model 2 (Madry et al., 2017) found after 1K iterations, with noise $L_2$-norm of 4.4. The third is one found after 10K iterations for Model 2, with noise $L_2$-norm of 2.1. The last image is the best attack on our Model 3 found after one million iterations, with noise $L_2$-norm of 3.5. These illustrates the trend shown in Table 1 that the defense by adversarial training diminishes as the attacks are allowed more iterations, while L2NNNs withstand strong attacks and it requires more noise to fool an L2NNN. It's worth noting that the slow degradation of Model 2's accuracy is an artifact of the attacker (Carlini & Wagner, 2017a): when gradients are near zero in some parts of the input space, which is true for MNIST Model 2 due to adversarial training, it takes more iterations to make progress. It is conceivable that, with a more advanced attacker, Model 2 could drop quickly to 7.6%. What truly matter are the robust accuracies where we advance the state of the art from 7.6% to 24.4%.

## C  DETAILS OF SCRAMBLED-LABEL EXPERIMENTS

For ordinary networks in Table 5, we use two network architectures. The first has 4 layers and is the architecture used in Madry et al. (2017). The second has 22 layers and is the architecture of Models 3 and 4 in Table 1, which includes norm-pooling and two-sided ReLU. Results of ordinary networks using these two architectures are in Tables 7 and 8 respectively. The ordinary-network section of Table 5 is entry-wise max of Tables 7 and 8.

In Tables 7 and 8, dropout rate and weight-decay weight are tuned for each WD/DR run, and each WD+DR+ES run uses the combined hyperparameters from its row. In early-stopping runs, 5000 training images are withheld as validation set and training stops when loss on validation set stops decreasing. Each ES or WD+DR+ES entry is an average over ten runs to account for randomness of the validation set. The L2NNNs do not use weight decay, dropout or early stopping.

Table 9 shows L2NNN trade-off points between accuracy and confidence gap on the 25%-scrambled training set. Table 10 shows L2NNN trade-off points between accuracy and confidence gap on

Table 7: Accuracies of non-L2NNN MNIST classifiers that use a 4-layer architecture and that are trained on training data with various amounts of scrambled labels. Rand is the percentage of training labels that are randomized. WD is weight decay. DR is dropout. ES is early stopping.

| Rand | Ordinary network | | | | |
| | Vanilla | WD | DR | ES | WD+DR+ES |
| --- | --- | --- | --- | --- | --- |
| 0 | 98.9% | 99.0% | 99.2% | 99.0% | 99.3% |
| 25% | 82.5% | 91.1% | 91.8% | 79.1% | 98.0% |
| 50% | 57.7% | 67.7% | 72.6% | 66.4% | 88.3% |
| 75% | 32.1% | 44.9% | 41.8% | 52.7% | 66.4% |
| 100% | 9.5% | 8.9% | 9.4% | NA | NA |

Table 8: Accuracies of non-L2NNN MNIST classifiers that use a 22-layer architecture and that are trained on training data with various amounts of scrambled labels. Rand is the percentage of training labels that are randomized. WD is weight decay. DR is dropout. ES is early stopping.

| Rand | Ordinary network | | | | |
| | Vanilla | WD | DR | ES | WD+DR+ES |
| --- | --- | --- | --- | --- | --- |
| 0 | 99.4% | 99.0% | 99.0% | 99.0% | 99.0% |
| 25% | 90.4% | 86.5% | 89.8% | 96.2% | 90.3% |
| 50% | 65.5% | 62.5% | 63.7% | 81.0% | 83.1% |
| 75% | 41.5% | 38.2% | 40.2% | 75.2% | 61.9% |
| 100% | 9.7% | 9.1% | 8.8% | NA | NA |

the 75%-scrambled training set. Like Table 6, they demonstrate the trade-off mechanism between memorization (training-set accuracy) and generalization (training-set average gap).

To be fair, dropout and early stopping are also able to sacrifice accuracy on a noisy training set. For example, the DR run in the 50%-scrambled row in Table 7 has 67.5% accuracy on the training set and 72.6% on the test set. However, the underlying mechanisms are very different from that of L2NNN. Dropout (Srivastava et al., 2014) has an effect of data augmentation, and, with a noisy training set, dropout can create a situation where the effective data complexity exceeds the network capacity. Therefore, the parameter training is stalled at a lowered accuracy on the training set, and we get better performance if the model tends to fit more of original labels and less of the scrambled labels. The mechanism of early stopping is straightforward and simply stops the training when it is mostly memorizing scrambled labels. We get better performance from early stopping if the parameter training tends to fit the original labels early. These mechanisms from dropout and early stopping are both brittle and may not allow parameter training enough opportunity to learn from the useful data points with original labels. The comparison in Table 5 suggests that they are inferior to L2NNN's trade-off mechanism as discussed in Section 3.3 and illustrated in Tables 6, 9 and 10. The L2NNNs in this paper do not use weight decay, dropout or early stopping, however it is conceivable that dropout may be complementary to L2NNNs.

## D PROOFS

**Lemma 1.** *Let $g(\mathbf{x})$ denote a single-L2NNN classifier's confidence gap for an input data point $\mathbf{x}$. The classifier will not change its answer as long as the input $\mathbf{x}$ is modified by no more than an $L_2$-norm of $g(\mathbf{x})/\sqrt{2}$.*

*Proof.* Let $\mathbf{y}(\mathbf{x}) = [y_1(\mathbf{x}), y_2(\mathbf{x}), \cdots, y_K(\mathbf{x})]$ denote logit vector of a single-L2NNN classifier for an input data point $\mathbf{x}$. Let $\mathbf{x_1}$ and $\mathbf{x_2}$ be two input vectors such that the classifier outputs different labels $i$ and $j$. By definitions, we have the following inequalities:

$$
\begin{aligned}
y_i(\mathbf{x_1}) - y_j(\mathbf{x_1}) &\geq g(\mathbf{x_1}) \\
y_i(\mathbf{x_2}) - y_j(\mathbf{x_2}) &\leq 0
\end{aligned}
\tag{12}
$$

Table 9: Training-accuracy-versus-confidence-gap trade-off points of L2NNNs on 25%-scrambled MNIST training labels.

| on training set | | on test set | |
|---|---|---|---|
| Accu. | Gap | Accu. | Gap |
| 99.6% | 0.12 | 92.6% | 0.10 |
| 97.6% | 0.20 | 95.7% | 0.17 |
| 78.6% | 0.31 | 98.2% | 0.30 |
| 77.2% | 0.64 | 98.5% | 0.63 |

Table 10: Training-accuracy-versus-confidence-gap trade-off points of L2NNNs on 75%-scrambled MNIST training labels.

| on training set | | on test set | |
|---|---|---|---|
| Accu. | Gap | Accu. | Gap |
| 97.9% | 0.07 | 49.8% | 0.03 |
| 93.0% | 0.09 | 59.2% | 0.05 |
| 75.9% | 0.10 | 70.0% | 0.08 |
| 58.0% | 0.18 | 80.4% | 0.17 |
| 46.2% | 0.29 | 86.8% | 0.30 |
| 40.1% | 0.44 | 89.8% | 0.46 |
| 34.7% | 0.86 | 93.1% | 0.89 |

Because the classifier is a single L2NNN, it must be true that:

$$
\begin{aligned}
\|\mathbf{x_2} - \mathbf{x_1}\|_2 &\geq \|\mathbf{y}(\mathbf{x_2}) - \mathbf{y}(\mathbf{x_1})\|_2 \\
&\geq \sqrt{(y_i(\mathbf{x_2}) - y_i(\mathbf{x_1}))^2 + (y_j(\mathbf{x_2}) - y_j(\mathbf{x_1}))^2} \\
&= \sqrt{(y_i(\mathbf{x_1}) - y_i(\mathbf{x_2}))^2 + (y_j(\mathbf{x_2}) - y_j(\mathbf{x_1}))^2} \\
&\geq \sqrt{\frac{(y_i(\mathbf{x_1}) - y_i(\mathbf{x_2}) + y_j(\mathbf{x_2}) - y_j(\mathbf{x_1}))^2}{2}} \\
&= \sqrt{\frac{((y_i(\mathbf{x_1}) - y_j(\mathbf{x_1})) + (y_j(\mathbf{x_2}) - y_i(\mathbf{x_2})))^2}{2}} \\
&\geq \sqrt{\frac{(g(\mathbf{x_1}) + 0)^2}{2}} \\
&= g(\mathbf{x_1})/\sqrt{2}
\end{aligned}
\tag{13}
$$

$\square$

**Lemma 2.** *Let $g(\mathbf{x})$ denote a classifier's confidence gap for an input data point $\mathbf{x}$. Let $d(\mathbf{x_1}, \mathbf{x_2})$ denote the $L_2$-distance between the output logit-vectors for two input points $\mathbf{x_1}$ and $\mathbf{x_2}$ that have different labels and that are classified correctly. Then this condition holds: $g(\mathbf{x_1}) + g(\mathbf{x_2}) \leq \sqrt{2} \cdot d(\mathbf{x_1}, \mathbf{x_2})$.*

*Proof.* Let $\mathbf{y}(\mathbf{x}) = [y_1(\mathbf{x}), y_2(\mathbf{x}), \cdots, y_K(\mathbf{x})]$ denote logit vector of a classifier for an input data point $\mathbf{x}$. Let $i$ and $j$ be the labels for $\mathbf{x_1}$ and $\mathbf{x_2}$. By definitions, we have the following inequalities:

$$
\begin{aligned}
y_i(\mathbf{x_1}) - y_j(\mathbf{x_1}) &\geq g(\mathbf{x_1}) \\
y_j(\mathbf{x_2}) - y_i(\mathbf{x_2}) &\geq g(\mathbf{x_2})
\end{aligned}
\tag{14}
$$

Therefore,

$$
\begin{aligned}
d\left(\mathbf{x_1}, \mathbf{x_2}\right) &\triangleq \left\|\mathbf{y}\left(\mathbf{x_2}\right) - \mathbf{y}\left(\mathbf{x_1}\right)\right\|_2 \\
&\geq \sqrt{\left(y_i\left(\mathbf{x_2}\right) - y_i\left(\mathbf{x_1}\right)\right)^2 + \left(y_j\left(\mathbf{x_2}\right) - y_j\left(\mathbf{x_1}\right)\right)^2} \\
&= \sqrt{\left(y_i\left(\mathbf{x_1}\right) - y_i\left(\mathbf{x_2}\right)\right)^2 + \left(y_j\left(\mathbf{x_2}\right) - y_j\left(\mathbf{x_1}\right)\right)^2} \\
&\geq \sqrt{\frac{\left(y_i\left(\mathbf{x_1}\right) - y_i\left(\mathbf{x_2}\right) + y_j\left(\mathbf{x_2}\right) - y_j\left(\mathbf{x_1}\right)\right)^2}{2}} \\
&= \sqrt{\frac{\left(\left(y_i\left(\mathbf{x_1}\right) - y_j\left(\mathbf{x_1}\right)\right) + \left(y_j\left(\mathbf{x_2}\right) - y_i\left(\mathbf{x_2}\right)\right)\right)^2}{2}} \\
&\geq \sqrt{\frac{\left(g\left(\mathbf{x_1}\right) + g\left(\mathbf{x_2}\right)\right)^2}{2}} \\
&= \frac{g\left(\mathbf{x_1}\right) + g\left(\mathbf{x_2}\right)}{\sqrt{2}}
\end{aligned}
\tag{15}
$$

$\square$

**Lemma 3.** *For any $a \geq 0$, $b \geq 0$, $p \geq 1$, the following inequality holds: $a^p + b^p \leq (a + b)^p$.*

*Proof.* If $a$ and $b$ are both zero, the inequality holds. If at least one of $a$ and $b$ is nonzero:

$$
\begin{aligned}
a^p + b^p &= (a+b)^p \cdot \left(\frac{a}{a+b}\right)^p + (a+b)^p \cdot \left(\frac{b}{a+b}\right)^p \\
&\leq (a+b)^p \cdot \frac{a}{a+b} + (a+b)^p \cdot \frac{b}{a+b} \\
&= (a+b)^p
\end{aligned}
\tag{16}
$$

$\square$

**Lemma 4.** *Let $f(x)$ be a nonexpansive and monotonically increasing scalar function. Define a function from $\mathbb{R}$ to $\mathbb{R}^2$: $\mathbf{h}(x) = [f(x), f(x) - x]$. Then $\mathbf{h}(x)$ is nonexpansive with respect to any $L_p$-norm.*

*Proof.* For any $x_1 > x_2$, by definition we have the following inequalities:

$$
\begin{aligned}
f(x_1) - f(x_2) &\geq 0 \\
f(x_1) - f(x_2) &\leq x_1 - x_2
\end{aligned}
\tag{17}
$$

For any $p \geq 1$, invoking Lemma 3 with $a = f(x_1) - f(x_2)$ and $b = x_1 - x_2 - f(x_1) + f(x_2)$, we have:

$$
\begin{aligned}
\left((f(x_1) - f(x_2))^p + (x_1 - x_2 - f(x_1) + f(x_2))^p \leq (x_1 - x_2)^p\right. \\
\left(((f(x_1) - f(x_2))^p + (x_1 - x_2 - f(x_1) + f(x_2))^p\right)^{1/p} \leq x_1 - x_2 \\
\left(|f(x_1) - f(x_2)|^p + |(f(x_1) - x_1) - (f(x_2) - x_2)|^p\right)^{1/p} \leq x_1 - x_2 \\
\|\mathbf{h}(x_1) - \mathbf{h}(x_2)\|_p \leq x_1 - x_2
\end{aligned}
\tag{18}
$$

$\square$

**Lemma 5.** *Norm-pooling within each pooling window is a nonexpansive map with respect to $L_2$-norm.*

*Proof.* Let $\mathbf{x_1}$ and $\mathbf{x_2}$ be two vectors with the size of a pooling window. By triangle inequality, we have

$$
\begin{aligned}
\|\mathbf{x_1} - \mathbf{x_2}\|_2 + \|\mathbf{x_1}\|_2 &\geq \|\mathbf{x_2}\|_2 \\
\|\mathbf{x_1} - \mathbf{x_2}\|_2 + \|\mathbf{x_2}\|_2 &\geq \|\mathbf{x_1}\|_2
\end{aligned}
\tag{19}
$$

Therefore,

$$\|\mathbf{x_1} - \mathbf{x_2}\|_2 \geq \|\mathbf{x_2}\|_2 - \|\mathbf{x_1}\|_2$$
$$\|\mathbf{x_1} - \mathbf{x_2}\|_2 \geq \|\mathbf{x_1}\|_2 - \|\mathbf{x_2}\|_2 \tag{20}$$

Therefore,

$$\|\mathbf{x_1} - \mathbf{x_2}\|_2 \geq |\|\mathbf{x_1}\|_2 - \|\mathbf{x_2}\|_2| \tag{21}$$

$\square$

**Lemma 6.** *Let $g(\mathbf{x})$ denote a multi-L2NNN classifier's confidence gap for an input data point $\mathbf{x}$. The classifier will not change its answer as long as the input $\mathbf{x}$ is modified by no more than an $L_2$-norm of $g(\mathbf{x})/2$.*

*Proof.* Let $\mathbf{y}(\mathbf{x}) = [y_1(\mathbf{x}), y_2(\mathbf{x}), \cdots, y_K(\mathbf{x})]$ denote logit vector of a multi-L2NNN classifier for an input data point $\mathbf{x}$. Let $\mathbf{x_1}$ and $\mathbf{x_2}$ be two input vectors such that the classifier outputs different labels $i$ and $j$. By definitions, we have the following inequalities:

$$y_i(\mathbf{x_1}) - y_j(\mathbf{x_1}) \geq g(\mathbf{x_1})$$
$$y_i(\mathbf{x_2}) - y_j(\mathbf{x_2}) \leq 0 \tag{22}$$

For a multi-L2NNN classifier, each logit is a nonexpansive function of the input, and it must be true that:

$$\|\mathbf{x_2} - \mathbf{x_1}\|_2 \geq |y_i(\mathbf{x_1}) - y_i(\mathbf{x_2})|$$
$$\|\mathbf{x_2} - \mathbf{x_1}\|_2 \geq |y_j(\mathbf{x_2}) - y_j(\mathbf{x_1})| \tag{23}$$

Therefore,

$$
\begin{aligned}
\|\mathbf{x_2} - \mathbf{x_1}\|_2 &\geq \frac{|y_i(\mathbf{x_1}) - y_i(\mathbf{x_2})| + |y_j(\mathbf{x_2}) - y_j(\mathbf{x_1})|}{2} \\
&\geq \frac{|y_i(\mathbf{x_1}) - y_i(\mathbf{x_2}) + y_j(\mathbf{x_2}) - y_j(\mathbf{x_1})|}{2} \\
&= \frac{|(y_i(\mathbf{x_1}) - y_j(\mathbf{x_1})) + (y_j(\mathbf{x_2}) - y_i(\mathbf{x_2}))|}{2} \\
&\geq \frac{|g(\mathbf{x_1}) + 0|}{2} \\
&= g(\mathbf{x_1})/2
\end{aligned}
\tag{24}
$$

$\square$

