# OpenReview forum: "L2-Nonexpansive Neural Networks"
_ICLR.cc/2019/Conference_

### Official Review · AnonReviewer1 · 2018-11-02
**The contribution of the method for combating adversarial examples does not look practically significant to me. Other contributions, like the ability to learn in the presence of label noise are more interesting, but require further development and experiments.**

**Rating:** 5
**Confidence:** 3

**Review:**

Summary:
The paper presents techniques for training a non expansive network, which keeps the Lipchitz constant of all layers lower than 1. While being non-expansive, means are taken to preserve distance information better than standard networks. The architectural changes required w.r.t standard networks are minor, and the most interesting changes are made to the loss minimized. The main claim of the paper is that the method is robust against adversarial attacks of a certain kind. However, the results presented show that a) such robustness comes at a high cost of accuracy for standard examples, and b) even though the network is preferable to a previous alternative in combating adversarial examples, the accuracy obtained in the face of adversarial attacks is too low to be of practical value. Other properties of the networks, explored empirically, are that the confidence of the prediction is indicative of robustness (to adversarial attacks) and that the networks learn better in the presence of high label noise.
In short, this paper may be of interest to a sub-community interested in defense against certain types of adversarial attacks, even when the defense level is much too low to be practical. I am not part of this community, hence did not find this part very interesting. I believe the regularization results are of wider interest. However, to present this as the main contribution of L2NNN more work is required to find configuration which are resilient to overfit yet enable high training accuracy, and more diverse experiments are required.
Pros:
+ the idea of non expansive network is interesting and important
+ results indicate some advantages in fighting adversarial examples and label noise
Cons:
- the results for fighting adversarial examples are not significant from a practical perspective
- the results for copying with label noise are preliminary and require expansion with more experiments.
- the method has costs in accuracy, which is lower than standard networks and this issue is not faced with enough attention
- presentation clarity is medium: proofs for claims are missing, as well as relevant background on the relevant adversarial attacks. The choice to place the related work at the end also reduces presentation clarity.

More detailed comments:
Pages 1-3: In many places, small proofs are left to the reader as ‘straightforward’. Examples are: the claim in the introduction, in eq. 2, in section 2.2, section 2.3’ last line of page 3, etc.. While the claim are true (in the cases I tried to verify them long enough), this makes reading difficult and not fluent. For some of these claims I do not see the argument behind them. In general, I think proofs should be brought for claims, and short proofs (preferably) should be brought for small claims. Leaving every proof to the reader as an exercise is not a convenient strategy.
Page 4: The loss is complex and its terms utility require empirical evidence. The third term is shown to be clearly useful, enabling a trade off between train accuracy and margin. However, the utility of terms 4) and 5) is not verified. Do we really need both these terms? Cannot we just stay with one?
The main claim is robustness w.r.t “white-box non targeted L2-bounded attacks”. This seems to be a very specific attack type, and it is not explained at all in the text. Hence it is hard to judge the value of this robustness. Explanation of adversarial attack kinds, and specifically of “white-box non targeted
L2-bounded attacks” is required for this paper to be a stand alone readable paper. Similarly ‘L_\infty’-bounded attacks, for which results are shown, should be explained.
Table 1,2: First, the model architecture used in these experiments is not stated. Second, the accuracy of the ‘natural’ baseline classifier, at least in the MNist case, is somewhat low – much better results can be obtained with CNN on MNist. Third, the accuracies of the suggested robust models are very low compared to what can be obtained on these datatsets. Forth, while the accuracies under attack of the proposed method are better than those of Madri et al., both are quite poor and indicate that the classifier is not useful under attack (from a practical perspective).
Page 6: The classifiers which share the work between an L2NNN network and a regular more accurate network may be interesting, as the accuracies reported for them are significantly higher than the L2NNN networks. However, the robustness scores are not reported for these classifiers, so it is not possible to judge if they lead to a practical and effective strategy.
Page 7: For me, the results with partially random labels are the most interesting in the paper. The resistance of L2NNN to overfit and its ability to learn with very noisy data are considerably better than the suggested alternatives.
Relevant work not mentioned “Spectral Norm Regularization for Improving the Generalizability of Deep Learning” - Yuichi Yoshida and Takeru Miyato, Arxiv, 2017.

I have read the rebuttal.
The discussion was interesting, but I do not see a need to change my assessment.
The example of ad-blocking in indeed a case (the first I encounter) where l2- perturbated adversarial examples can be useful for cyber attack. The other ones are less relevant (the attacks are not based on adversarial attacks in the sense used in the paper: images crated with small gradient-direction perturbations). Anyway talking about 'attacks on a self-driving car' are still not neaningful to me: I do not understand what adversarial examples have to do with this.
I do not find the analogy of 'rocket improvements and moon landing' convincing: in 69 rocket improvements were of high interest in multiple applications, and moon landing was visible over the corner.

---

> ### Author Response · Authors · 2018-11-10
> **our responses, part 2 of 2**
>
> 3) Regarding the robustness-accuracy trade-off.
> There is indeed a robustness versus nominal accuracy trade-off. We reported the trade-off in the second to last paragraph of Section 3.1, and we revisited the topic in the second to last paragraph of Section 3.3 to state a hypothesis.
> We are not the only defense work that face this trade-off. As AnonReviewer2 pointed out, adversarial training has a similar trade-off. It also can be seen in the adversarial polytope work of https://arxiv.org/abs/1805.12514. It remains an open question whether such trade-off is a necessary part of life.
> Our hypothesis on L2NNN's trade-off is stated at end of Section 3.3: by having a second goal of enlarging confidence gap, L2NNN's parameter training automatically and selectively misclassify certain training data in exchange for larger gaps. In the context of original training labels, this implies that some original labels are ignored and that leads to lower nominal accuracy. Although by looking at examples in Figure 4, one could argue that some of the original labels are better ignored. This is only a hypothesis. We agree with the reviewer that this trade-off is an important subject to study, we may have a better answer in future work.
>
> 4) Regarding omitted proofs.
> We thank the reviewer for the suggestions and we are updating the appendix to add proofs, and will post the revision soon.
>
> 5) Regarding loss terms (4) and (5).
> Removing one of these two terms would not result in as much degradation as in Table 3. If to choose one of the two, it makes sense to use (5) and the end result would be slight degradation in nominal accuracy compared with the current results in Table 1 and 2.
>
> 6) Regarding architecture.
> Our models 3 and 4 in Table 1 and 2 all use convolution layers followed by fully connected layers, some of which are split layers with stacks dedicated to individual logits. These are conventional architecture choices, and our unconventional elements are two-sided ReLU and norm-pooling. By the way, they are all available for download at the dropbox link on page 4.
> For the scrambled-label experiments, as detailed in Tables 7 and 8, we want to be fair and build ordinary networks with two different architectures, one shallow and one deep. Then the ordinary-network section of Table 5 is entry-wise max of Tables 7 and 8. The L2NNNs use the same architecture for MNIST throughout this paper.
>
> 7) Regarding hybrid models reported at end of Section 3.2.
> The following are measurements of the said hybrid models under same settings in Tables 1 and 2, after 1000 iterations: MNIST 62.9%, CIFAR-10 6.4%. Please note that the base for comparison is Models 3 in Tables 1 and 2. The CIFAR-10 number is in line with expectation, and 6.4% is a degradation from 10.1%. The MNIST number, however, is an artifact of that the CW attack code was not designed for ensemble models and Carlini & Wagner can likely do much better if they know and take advantage of the hybrid mechanism. The real MNIST number ought to be somewhat below 20.1%.
>
> 8) Regarding missing reference.
> Thank you and we will added the reference.
>
> Please let us know if we have missed anything and we'd be happy to continue the discussion.

---

> > ### Comment · AnonReviewer1 · 2018-11-11
> > **Is a (completely impractical)  defense against adversarial attacks  a good topic for a top-notch paper?**
> >
> > I think the answer is 'no', and this is the main reason I think the paper should be rejected.
> >
> > The problem is two folded:
> > 1) I do not think adversarial attacks a real threat on AI systems.
> > I did not see here, or in any other paper, a presentation of a convincing scenario in which adversarial attacks can be used to cause harm which cannot be caused with much simpler means. Recall that adversarial attacks are creating images which look of a certain class to humans, yet a different class to machines. Where and how can these examples be used for cyber attack purposes? How are they a threat?
> > Moreover, white box examples require acquaintance with the model to produce - an even harder task for the attacker.
> > For example: the authors say "It is absolutely true that we cannot put a 24.4%-robust classifier in a self-driving car and declare mission accomplished"  - before agreeing that 24.4% is not good enough, one should first argue that one can attack a self-driving car with adversarial attacks. I do not currently see how: if someone malicious has gained access to the camera of a self driving car, why would he want to present it with adversarial examples? instead he can just feed it with images of a fake reality and cause harm directly.
> >
> > I would be glad to hear arguments/examples showing that adversarial examples are of cyber importance or of any practical importance. This may change my view of the topic and the paper.
> >
> > 2) the second point was already mentioned in the previous discussion: if adversarial examples could be used to cause harm, the current method would not help. It just does not defend well enough (as well as all other methods).
> >
> > If such a paper is to be accepted, I think a thorough discussion is required in its introduction to explain why adversarial examples are of interest at all. This is far from being clear.

---

> > > ### Author Response · Authors · 2018-11-11
> > > **three examples**
> > >
> > > Please consider the following three examples.
> > >
> > > The first example is from last week:
> > > https://arxiv.org/abs/1811.03194
> > > It demonstrated that ad-blocking based on neural networks is vulnerable and easily defeated by adversarial examples. Furthermore, if ad-blocking based on neural networks is deployed, it "would engender a new arms race that overwhelmingly favors publishers and ad-networks." Also because ad-blocking based on neural networks needs to run with a high privilege level inside a browser, it would "introduce new vulnerabilities that let an attacker bypass web security boundaries and mount DDoS attacks."
> > >
> > > The second example is:
> > > https://www.cs.cmu.edu/~sbhagava/papers/face-rec-ccs16.pdf
> > > It demonstrated that by printing eyeglass frames, one person can impersonate another person (a specific choice) in front of a state-of-the-art face-recognition algorithm. Please see their figure 4, the authors can pretend to be Milla Jovovich and Carson Daly by just wearing glasses.
> > >
> > > The third example is from CVPR 2018:
> > > https://arxiv.org/pdf/1707.08945.pdf
> > > It demonstrated that by putting just four stickers on a stop sign, a neural network would recognize it as a "speed limit 45" sign, and it would make the same mistake consistently from different distances and angles. The authors also performed field test in a moving vehicle.
> > >
> > > As we mentioned, currently the attack-side research has an upper hand over the defense side, and they are only getting better and entering the physical world more and more. It is the more reason to encourage research on adversarial defense.
> > >
> > > People in the security field are concerned enough that a security conference accepted Carlini & Wagner (2017a) and gave it best student paper award, and it has received more than 500 citations by today. ICLR 2018 accepted Madry et al. (2017) which has received more than 250 citations.
> > >
> > > Regarding the reviewer's second point. Allow us use an analogy: before 1969, the scientific community should not reject paper on rocket improvements on the ground that nobody had landed on the moon.

---

> > > > ### Public Comment · ~Justin_Gilmer1 · 2018-11-13
> > > > **Consider evaluating on out-of-distribution inputs**
> > > >
> > > > The examples 1 and 3 you are giving of adversarial examples for real world systems only explore small perturbations without much justification for doing so. For the ad-block case, the attacker has full control of the image and can design any image from scratch that does not match the models' hash. Why should the attacker restrict themselves to small perturbations of a correctly handled image? For street signs an attacker could knock it over, construct a large adversarial “yard sale” sign placed next to the sign, or just place a bag over it. Given how brittle classifiers are on out-of-distribution inputs, it’s plausible that some signs may be misclassified just because it’s a foggy day. Example 2 was discussed in https://arxiv.org/abs/1807.06732 . Glasses won’t be useful in settings where a security guard can match the person’s face to the face returned by the image recognition system (they could also be instructed to have people remove all accessories before being scanned). Even for some settings where no guard is present and checking the model's output, small l_2 perturbations do not capture the action space of the attacker.
> > > >
> > > > It's clear the adversarial example field is popular and receives many citations, but that doesn’t mean on its own that this is well motivated security research. What some critics are asking for is that security-motivated work perform a realistic assessment of the threats facing actual systems that is unbiased by the "surprising" phenomenon of small adversarial perturbations. The papers you cite seem more interested in attacking systems with small perturbations rather than properly designing a threat model that considers all of the options available to the attacker. There is a clear mismatch between how real world attackers typically break actual systems and what the adversarial example field currently focuses on. Consider for example the adversarial image spam shown in Figure 2 of https://arxiv.org/pdf/1712.03141.pdf, this is what attackers actually do. Anyone who has browsed Youtube has probably encountered adversarially modified videos that have been uploaded to evade copyright detection, such modifications typically involve very large and obvious transformations to the input.
> > > >
> > > > Your evaluation only demonstrates small improvements to a toy threat model. Can you provide evidence that your method is progress towards securing systems in more realistic threat models? If you can demonstrate that your method improves model generalization on out-of-distribution inputs then that could be more convincing. For example, the madry adversarial defense was 0% robust to the “lines” attack defined in Section 5.1.1 of https://arxiv.org/abs/1807.06732, does your method improve robustness to this unseen transformation? Even better, one could evaluate on the recently proposed common corruption benchmark https://openreview.net/forum?id=HJz6tiCqYm, which considers a host of realistic image corruptions that models may face at deployment.

---

> > > > > ### Public Comment · (anonymous) · 2018-11-14
> > > > > **Indeed**
> > > > >
> > > > > Such high ratings and evaluations for papers on adversarial robustness to L_2 or L_inf attacks which have a completely unrealistic threat model while other more practical papers with real world impact struggle, seems completely unreasonable to me.

---

> > > > > ### Author Response · Authors · 2018-11-14
> > > > > **we must learn to walk before we can run. (with apologies to Area Chair)**
> > > > >
> > > > > Hi Justin,
> > > > >
> > > > > On the first example of ad-blocking.
> > > > > An ad publisher has a content that he/she wants to deliver. That content, if without perturbation, would be correctly handled, i.e. blocked, by perceptual ad-blocking. Hence he/she has a motivation to deliver that content by adding small perturbations.
> > > > > Humans are good at filtering out ads, and a spammer who uses the lower image in Figure 2 of https://arxiv.org/pdf/1712.03141.pdf would be an unsuccessful spammer, because nobody would pay attention to contents in such an image. A successful spammer is one that can take the upper image, add small perturbations, and deliver it to people's inboxes. For phishing and spam attacks it's even more important that people can not recognize them as such, and imperceptible perturbations are useful.
> > > > >
> > > > > On the second example of face recognition.
> > > > > Nowadays many people use their faces to unlock cellphones. If someone can print eyeglass frames to gain access to other people's devices, it's a big security gap for millions of people.
> > > > > By the way, the relative magnitude of an eyeglass frame with respect to a face is roughly on par with the relative magnitude of L2 norm of 3 with respect to an MNIST image.
> > > > >
> > > > > On the third example of stop sign.
> > > > > If a stop sign gets knocked over or covered, a policeman will correct the situation. If a stop sign has four stickers on it, nobody would bother until an accident happens.
> > > > > By the way, the relative magnitude of those four stickers with respect to the whole stop sign is roughly on par with the relative magnitude of L2 norm of 3 with respect to an MNIST image.
> > > > >
> > > > > Some of Justin's arguments seem to be that L2 and L_inf norm metrics do not constitute a sufficient condition for a robust classifier. That is absolutely true. For example, if one has a MNIST classifier that is 90% robust against L2 norm of 3 and L_inf norm of 0.3, it might still break down if an input image is rotated by an angle, while a truly robust classifier like a human would not change its decision. By the way, the "lines" attack and common corruption benchmark do not constitute a sufficient condition either, and the lines-attack pictures on Madry et al. (2017) seem more excusable than our Figure 5 (nonetheless we'll be happy to evaluate our models under those conditions).
> > > > >
> > > > > The flip-side question is whether robustness as measured by L2 and/or L_inf norm metrics is a necessary condition for a robust classifier. The answer is a big yes. We must learn to walk before we can run, and frankly the status quo of neural-network robustness research is at crawling stage. While we are crawling, we do not have the luxury to look down upon people who are working to meet necessary conditions, and rejecting defense papers based on an argument of not having a sufficient condition would only hinder progress and make those more hefty goals the more difficult to reach.
> > > > >
> > > > > We agree with many points Justin made in this paper
> > > > > https://arxiv.org/pdf/1807.06732.pdf
> > > > > However, if the conclusion is to suppress the "perturbation defense" literature, that would be wrong. In our opinion, advances in white-box defense, a.k.a. robustness, a.k.a. generalization, of neural networks as measured by L2 norm and/or L_inf norm metrics are not only valid but also essential topics for the deep learning community, and our paper represents a big step forward. Given the size of improvement, L2NNN is likely a key ingredient in future truly robust solutions, which people have little clue yet. When people do find those solutions, there would be profound impacts across the board on both security and generalization.

---

> > > > > > ### Public Comment · ~Justin_Gilmer1 · 2018-11-16
> > > > > > **I'm not trying to suppress the defense literature, I'm trying to improve it**
> > > > > >
> > > > > > First off, thanks to everyone for the thoughtful discussion. I appreciate the area chair’s commitment to evaluating the technical details of the methods proposed in this paper, and I am not passing any judgement on the specifics of this proposed method. I would also like to apologize to the authors, I am trying to have a broader discussion about how this field can have a larger impact rather than trying to get any paper rejected. I imagine it must be frustrating to feel like you suddenly need to defend the motivation of your work when many similar prior works have been published, and a decent subset of the ML community currently finds lp robustness in itself an interesting topic of study. However, I do hope that future researchers interested in adversarial robustness consider my recommendation and begin evaluating on more general out-of-distribution inputs, at least in addition to lp robustness. Doing so can only help increase the impact of their work --- it will make their work interesting to a larger subset of researchers (and potentially satisfy reviewers who don’t find lp robustness interesting), and will help broaden our understanding of model performance in non iid settings. Plus, it will have the added benefit that if later researchers falsify your lp-robustness numbers you will still have an irrefutable evaluation to fall back on. Researchers are of course welcome to report lp robustness if they find it interesting, but in my opinion by remaining fixated on small worst-case perturbations we are missing an opportunity to better understand and improve model performance in more general and realistic settings. Just as the authors identify that l2 robustness is a necessary step towards secure systems, so is achieving 0 test error with respect to distributional shift. It also seems likely that defenses which are truly making progress on this problem will also show improved robustness to distributional shift, indeed it has already been shown that adversarial training does help improve model robustness to a host of more general image corruptions (see Section 5.2 of https://openreview.net/pdf?id=HJz6tiCqYm ).
> > > > > > 	Regarding the security motivation. Reviewer2 correctly points out that most ML researchers (myself included) are not security researchers, and many of us have not spent the time thinking seriously about threat modeling. I have had the benefit of having discussions on this topic with security researchers as well as consulting with a product team over the threats their specific system faces, and doing so has changed how I view adversarial robustness. What I can say is that for most systems I’m aware of, I would not recommend that practitioners deploy a method that increases test error while improving lp robustness. Doing so would only reduce the security of the system. In the case of an ad-block system, increasing test error would increase the rate at which non-adversarially modified ads get past your system. For the small subset of sophisticated attackers, the gains in lp robustness would only force them to be slightly more clever in the modifications they make. If their company logo is an elephant, perhaps they could try shifting the location of the logo in their ad, similar modifications have been shown to cause model errors for object detection systems https://arxiv.org/abs/1808.03305. I would recommend the introduction of this paper have a more nuanced discussion around the security motivation of small adversarial perturbations. If there is a system for which we feel like this threat model is relevant, be specific and explain why. If your method increases test error, explain why that is tolerable for the considered application.
> > > > > >      Thanks again for the discussion everyone. Overall, I want the same thing that adversarial example researchers want, which is more secure and robust models. I will be at NIPS and would be excited to discuss this topic in detail with anyone in person!

---

> > > > > > > ### Author Response · Authors · 2018-11-16
> > > > > > > **let's agree to disagree**
> > > > > > >
> > > > > > > We do not mind at all that this debate happens at our paper's page, and if people wish to continue the discussion please do.
> > > > > > > However we the authors will stop responding to this particular subject, heeding the sage words of the Area Chair.

---

> > > > ### Comment · Area_Chair1 · 2018-11-14
> > > > **Discussion of technical points**
> > > >
> > > > Thanks to the authors and to the reviewers and various other commenters for the vigorous discussion surrounding the L2 threat model. While this can no doubt be discussed at great length (and all participants should feel free to do so), I want to note that I am most interested in judgments regarding the specific technical content of this paper. It is clear that there are both a number of researchers excited about exploring the Lp-norm threat model, and a number who are deeply skeptical of the model. I am very familiar with the arguments in both directions and will take both viewpoints into account in my eventual meta-review.
> > > >
> > > > So by all means continue discussing, but consider turning your energy towards the specifics of the paper rather than whether this general area is worth exploring. This has the greatest chance of influencing the eventual decision.
> > > >
> > > > Best,
> > > > ICLR Area Chair

---

> > > ### Comment · AnonReviewer2 · 2018-11-14
> > > **Yes, adversarial machine learning is interesting**
> > >
> > > First of all, adversarial examples give us insights into the limitations of current machine learning methods. This is valuable even if there are no practical attacks that represent an immediate security threat.
> > >
> > > Second, adversarial training methods allow us to enforce additional constraints on how we generalize from data. In some cases (e.g., computer vision), we have reason to believe that small perturbations of the input should not change the predicted label. This is another kind of background knowledge that can improve generalization.
> > >
> > > Third, there are indeed potential security risks! Here's a recent example:
> > >   https://www.cs.dartmouth.edu/farid/downloads/publications/eusipco18.pdf
> > >
> > > Summary: People who manipulate images can disguise their manipulations by printing out the image and taking a picture of it. The result is a real photograph of a manipulated photograph. Convolutional neural networks can be trained to detect these attacks, but they can also be fooled by small perturbations to the input. Extensive retraining reduces this vulnerability somewhat.
> > >
> > > The reason why most adversarial machine learning work doesn't have a good threat model is that most machine learning researchers are not security researchers. But that doesn't mean there aren't risks, and it doesn't make the work useless.

---

> > > > ### Author Response · Authors · 2018-11-14
> > > > **thank you very much**
> > > >
> > > > We strongly agree with AnonReviewer2's arguments.

---

> ### Author Response · Authors · 2018-11-10
> **our responses, part 1 of 2**
>
> We thank the reviewer for the thoughtful review and helpful suggestions, especially for appreciating the scrambled-label experiments. We are updating the paper to incorporate some of the suggestions and will post a revision soon. In the responses below, related points are grouped together and ordered roughly by significance.
>
> Before going into the list, we wish to emphasize that this paper sets a new state of the art in adversarial defense. Currently in the field, the attack side has an upper hand over the defense side, and indeed there has not been a defense that is practically significant, as the reviewer put it, from the perspective of real-life applications. However, if and when that happens, there will be wide implications across most deep-learning applications in terms of both security and generalization. That is more reason to look for advances in defense, and our paper represents a big step forward.
>
> 1) Regarding the significance of our defense results and types of attacks.
> Let us put our defense results in context. The white-box non-targeted scenario is the easiest for attacker and the most difficult for defense. White-box means that the attacker has complete information of a classifier, i.e. its architecture and parameters. By definition, if a classifier achieves a certain degree of white-box defense, its defense in black-box or transfer-attack scenarios can only be higher. Non-targeted means that any misclassification is considered a successful attack, while targeted attacks must reach a certain label. If a classifier achieves a certain degree of defense against non-targeted attacks, its defense in targeted scenario can only be higher. Therefore, white-box non-targeted defense is the holy grail of defense research, as it subsumes other types, and that's what we focus on.
> Then there is the choice of how to quantify noise, and the consensus in the field seems to be L2 norm or L_inf norm, preferably both. This choice leads to two measurements, defense against L2-bounded attacks and defense against L_inf-bounded attacks.
> White-box defense has been an elusive goal and numerous defense proposals have failed. Before our work, adversarial training has been considered the mainstream approach and Madry et al. (2017) has been the state of the art.
> This paper sets a new state of the art for defense against white-box non-targeted L2-bounded attacks. The reviewer commented that it is a very specific attack type, and we want to point out this type subsumes all other L2-bounded types. At the same time, L2NNNs also exhibit, in Table 4, near-state-of-the-art defense against white-box non-targeted L_inf-bounded attacks, which subsumes all other L_inf-bounded types. It is absolutely true that we cannot put a 24.4%-robust classifier in a self-driving car and declare mission accomplished. However, L2NNNs produce better defense than all other methods that are known to the field, and our results point to a different direction than what people thought as the mainstream approach.
> The degree of interest in our results can be felt by the number of non-reviewer comments we get, and one commenter has kindly tested our models, please see the comment titled "Very well done evaluation". We argue that that is side evidence that our results represents meaningful development.
> We agree with the reviewer that a better introduction would make this paper more accessible to readers outside the subfield of adversarial attack and defense, and we will improve on that if this paper gets accepted and more space is allowed in final version.
>
> 2) Regarding L2NNN as a general regularization technique beyond adversarial defense.
> We thank the reviewer for the appreciation, and we ourselves are proud of our scrambled-label results. The results also provide a partial answer to the questions posed by Zhang et al. (2017) (best paper award ICLR 2017) which reported that no traditional regularization techniques seem to stop neural networks from memorizing random labels. Our results suggest that L2NNN is one regularization technique that can suppress memorization in exchange for stronger generalization.
> We agree strongly with the reviewer that L2NNN as regularization has wider potentials outside adversarial defense. This indeed warrants a comprehensive study, which we will pursue in future works. For this paper, adversarial defense is our main results, and we want to scratch the surface for L2NNN's other properties.

---

### Official Review · AnonReviewer3 · 2018-11-02
**Nice idea, a few cool results, and a couple missing steps**

**Rating:** 6
**Confidence:** 4

**Review:**

I read this paper with some excitement. The authors propose a very sensible idea: simultaneously maximizing the confidence gap and constraining the Lipschitz constant of the network, thus achieving a guarantee that no L2-bounded perturbation can alter the prediction so long as the perturbation is bounded by some function of the confidence gap.

The main idea consists of three parts:
 1) smooth networks (fixed, low Lipschitz constant)
 (2) loss function that explicitly maximizes the confidence gap (distance between largest and second-largest logits).
 (3) “the network architecture restricts confidence gaps as little as possible. We will elaborate.”

The first two conditions make plain sense. The third condition and subsequent elaborations are far too vague. What precisely is the property of restricting confidence gaps? At first glance this seems akin to the smoothness sought in property one. Even in the bulleted list, the authors owe the reader a clearer explanation.

The proposed model, denoted L2-nonexpansive neural networks (L2NNNs) and consists of a sensible form of Lipschitz-constant-enforcing weight regularization, a loss function that penalizes the confidence gap.

To address the third condition, the authors say only “we adapt various layers in new ways for the third condition, for example norm-pooling and two-sided ReLU, which will be presented later” which is far too vacuous. At this point the reader is exposed to the third condition for the second time and yet it remains shrouded in mystery. The authors should elaborate here and describe what precisely, if anything, this third condition consists of. If it is not rigorously defined but only a heuristic notion, that would be fine, but this should be communicated clearly to the reader.

A following paragraph introduces the notion of “preserving distance”. However, what follows is too informal a discussion, and the rigorous definition never materializes. The authors say in one place “a network that maximizes confidence gaps well must be one that preserves distance well”. In this case, why do we need the third condition at all if the second condition appears to be sufficient?

In the next sections the authors describe the methods in greater detail and summarize their results. I have placed some more specific nittier comments in the ***small issues*** section below. But comment hear on the empirical findings.

One undersold finding here is that the existing methods (including the widely-believed-to-be-robust method due to MAdry 2017) that appear robust under FGSM attacks break badly under iterated attacks, and that the attacks go stronger up to 1M iterations, bringing accuracy below 10%.

In contrast the proposed method reaches 24% accuracy, which isn’t magnificent, but does appear to outperform the model due to Madry. A comparison against the method due to Kolter & Wong seems in order. The authors do not implement methods based on the adversarial polytope due to their present un-scalability, but that argument would be better supported if the authors were addressing larger models on harder datasets (vs MNIST and CIFAR10).

In short, I like the main ideas in this paper although some more empirical elbow grease is in order, the third condition needs to be discussed more rigorously or discarded. Additionally the choice of loss function should be better justified. Why do we need the original cross-entropy objective at all. Why not directly optimize the confidence gap? Did the authors try this? Did it work? Apologies if I missed this detail. Overall, I am interested in the development of this paper and would like to give it a higher vote but believe the authors have a bit more work to do to make this an easier decision. Looking forward to reading the rebuttal.


***Small issues***
Page 1 “nonexpansive neural networks (L2NNN)” for agreement on pluralization, should be “L2NNNs”

“They generalize better from noisy training labels than ordinary networks: for example, when 75% of MNIST training labels are randomized, an L2NNN still achieves 93.1% accuracy on the test set”
When you make a claim about accuracy of a proposed model, it must be made in reference to a standard model, even in the intro. It’s well-known in general that DNNs perform well even under large amounts of label noise. Hard to say without reference if 93.1% represents a significant improvement.

Repeated phrase on page 2:
“How to adapt subtleties like recursion and splitting-reconvergence is included in the appendix.”
“Discussions on splitting-reconvergence, recursion and normalization are in the appendix.”

Inputs to softmax cross-entropy should be both a set of logits and the label -- here the way the function is used in the notation does not match the proper function signature

Figure --- do not put “Model1, Model2, Model3, Model4”. This is unreadable. Put some shortname and then define it in the caption. Once one knows the abbreviations, they should be able to look at the figure and understand it without constantly referencing the caption.

Table 1-4 should be at the top of the page and arranged in a grid.  This wrapfigure floating in the middle of the page, while purely a cosmetic issue that should not bear on our deliberations, tortures the template unnecessarily, turning the middle 80% of page 5 into a one-column page unnecessarily.

Table 4 should show comparison to Madry model. Also this is why you need a shortname in the legend. In order to understand table 4, the reader has to consult the caption for tables 1 and 2.

“It is an important property that an L2NNN has an easily accessible measurement on how robust its decisions are”
I AGREE!

---

> ### Author Response · Authors · 2018-11-10
> **our responses, part 2 of 2**
>
>
> 4) Regarding presentation issues.
> Thank you very much for the suggestions and we agree with most. We are making some of the changes and will post a revision soon, and will do more if this paper is accepted and more space is allowed in final version.
> One thing we want to point out that that L2NNN's 93.1% performance from 75%-random training labels is significantly higher than the best of ordinary networks, see Tables 5,7,8.
> Our measurements of L_inf defense of Madry et al. (2017) are close to those reported in their paper. Because we use the same L_inf epsilon values, the numbers in Table 4 can be directly compared against those reported in Madry et al. (2017), Kolter & Wong (2017) and Raghunathan et al. (2018). As we acknowledged at end of section 3.1, Madry et al.'s MNIST L_inf result is still the best, while for CIFAR-10 we are on par.
>
> 5) We agree with the reviewer that pointing out that MNIST is not a solved problem is important to the field as well.
> And we thank the reviewer for appreciating that L2NNNs have an easily accessible measure of robustness.
>
> Please let us know if we have missed anything and we'd be happy to continue the discussion.

---

> > ### Comment · AnonReviewer3 · 2018-12-05
> > **Thanks**
> >
> > Thanks for the thorough replies. I will keep these clarifications and updates in mind as I reassess the paper and discuss with other reviewers.

---

> ### Author Response · Authors · 2018-11-10
> **our responses, part 1 of 2**
>
> We thank the reviewer for the thoughtful review and many helpful suggestions. We are updating the paper to incorporate some of the suggestions and will post a revision soon. In the responses below, related points are grouped together and ordered roughly by significance.
>
> 1) Regarding the third condition (preserving distance).
> We thank the reviewer for pointing out presentation issues related to the third condition.
> In short, the second and third conditions are two aspects of enlarging confidence gaps: the second condition does so by modifying the loss function, while the third condition does so by modifying the network architecture. The practical embodiments of the third condition include two-sided ReLU, norm-pooling, and a few more in appendix A.
> It is true that we did not derive two-sided ReLU or norm-pooling from a mathematical formulation of the third condition. Rather we started from a heuristic notion, as the reviewer put it, of preserving distance, and hypothesized that popular non-linear functions like ReLU and max-pooling unnecessarily restrict confidence gaps (see Sections 2.2 and 2.3), and hypothesized about two-sided ReLU and norm-pooling as improvements on preserving distance, and empirically verified their effects in enlarging confidence gaps and improving robustness.
> The argument that "a network that maximizes confidence gaps well must be one that preserves distance well" was meant to say that preserving-distance-well is a necessary property for an L2NNN with large confidence gaps, and hence motivate architecture changes. Again the second condition is about the loss function and the third condition is about architecture.
> Results in Table 3 suggest that architecture choices are important for nonexpansive networks, specifically that some non-linear functions that are not in common practice work better than the more standard ones. These unusual functions, two-sided ReLU and norm-pooling, have the property that they do a better job at preserving distance and let the parameter training, rather than architecture, determine what information is thrown away. Empirical results support that these functions are best practice when used in nonexpansive networks. In contrast, preserving distance is less important in ordinary networks because parameter training can choose weights that amplify distances arbitrarily. There are likely more architecture choices for nonexpansive networks which may improve future results, and architecture exploration is one of our future directions.
>
> 2) Regarding comparison against Kolter & Wong (2017).
> We did not realize that the Kolter & Wong (2017) models are available for download. The mention of scalability in Section 4 was part of literature review and not intended as an excuse. We can certainly try and put the Kolter & Wong (2017) models through the same L2-defense comparisons as in Tables 1 and 2. This may take some time as we need to port the models to be compatible with the CW attack code, and we will report back here.
> For L_inf defense, we did report a comparison. Table 4 shows our L_inf defense results under the same epsilon 0.1 as used in Kolter & Wong (2017). The results suggest that the measured L_inf defense is roughly on par.
> We will also add reference to their follow-up paper: https://arxiv.org/abs/1805.12514.
>
> 3) Regarding using only the confidence-gap loss.
> Unfortunately the confidence-gap loss (6) alone would not work. The problem is that (6) is too weak in penalizing mistakes. Consider a hypothetical MNIST neural network that always outputs logit value 1000 for label 0, and logit value 0 for the other nine labels. It would be a useless classifier, yet its (6) loss would be approximately -100 (-1000*0.1+0*0.9), which is lower than a useful classifier.
> The reviewer might be interested to see what happens if we put more weight on the confidence-gap loss. We reported additional accuracy-robustness trade-off points in the second to last paragraph of Section 3.1. That trade-off curve continues and here is another point with heavier weight on (6): nominal accuracy drops to 97.9% and the robust accuracy (1000-iteration attacks) increases to 24.7%. By the way, in the second to last paragraph of Section 3.3 we stated a hypothesis that this tradeoff is due to fundamentally the same mechanism as the tradeoff shown in Table 6.

---

### Official Review · AnonReviewer2 · 2018-11-05
**A variety of methods combine to give L2-robustness**

**Rating:** 8
**Confidence:** 4

**Review:**

This paper presents a combination of methods that, together, yield neural networks that are robust to small changes in L2 distance. The main idea is to ensure that changing the input by a bounded L2 distance never changes the output by more than the same L2 distance. Then, the difference between the highest-scoring class and the second-highest scoring class provides a bound on how much the input must change. The trivial way to do this is to rescale the final output layer so that all of the magnitudes are very small; however, this would give no additional robustness at all. To counteract this, the paper introduces several additional heuristics for increasing the gap between the highest-scoring class and the second-highest scoring one. Adversarial training can be used to make the models even more robust.

Experimental results on MNIST and CIFAR look impressive, although most are in terms of L2 distance, while most previous work optimizes L_infinity distance.

The methods described by this paper are similar to max-margin training, which is already known to be optimally robust to L2 perturbations for linear models (e.g., Xu et al. (2009)). This paper would be stronger with more discussion and analysis of this connection, although that might be work for a future paper.

Although the method relies heavily on heuristics, the empirical results are promising. The analysis of the contribution of the heuristics is fairly thorough as well. The MNIST results are strong. The CIFAR results show improved robustness, though at reduced accuracy on natural images. A combination of robust and non-robust classifiers improves the accuracy somewhat.

Overall, this is interesting work with promising empirical results. The biggest weaknesses are:

- Limited theory. The loss function is particularly strange.

- The majority of the comparisons focus on L2-robustness, but are comparing to a model optimized for L_infinity-robustness. (Thankfully, the authors also do some comparisons on L_infinity-robustness.)

- Robustness comes at a cost in accuracy, though this is not uncommon for adversarial training.

The biggest strengths are:

- Strong empirical robustness

- Analysis of combinations of methods and their interactions: different loss function, different architecture, different weight constraints, and adversarial training are all evaluated together and separately.

- Wide variety of experiments, including generalization on training data with noisy labels and analysis of the confidence gaps.



Questions for the authors:

- For equation (4) in the loss function, why would rescaling the layers in the middle of the network be equivalent to a linear transformation (u1, u2, ..., u_K) of the output?

- In equation (6), what is the average averaging over?

- The connection between confidence gap and robustness is discussed empirically, as a correlation, rather than theoretically, as a bound.  Doesn't the confidence gap give a lower bound on the minimum perturbation to change the predicted class?

---------

EDIT: After the author response, I remain positive about this paper. In addition to addressing my concerns, I admire the authors' patience in answering the concerns of other reviewers and commenters. I think that this is a solid paper that makes a good contribution to the literature on adversarial machine learning.

---

> ### Author Response · Authors · 2018-11-10
> **our responses, part 2 of 2**
>
> 4) Regarding the robustness-accuracy tradeoff.
> There is indeed a robustness versus nominal accuracy trade-off. We reported the trade-off in the second to last paragraph of Section 3.1, and we revisited the topic in the second to last paragraph of Section 3.3 to state a hypothesis.
> As the reviewer pointed out, we are not the only defense work that face this trade-off. It remains an open question whether such trade-off is a necessary part of life.
> Our hypothesis on L2NNN's trade-off is stated at end of Section 3.3: by having a second goal of enlarging confidence gaps, L2NNN's parameter training automatically and selectively misclassify certain training data in exchange for larger gaps. In the context of original training labels, this implies that some original labels are ignored and that leads to lower nominal accuracy. Although by looking at examples in Figure 4, one could argue that some of the original labels are better ignored. If this hypothesis is true, this trade-off mechanism is a double-edged sword, as it both costs us nominal accuracy in Tables 1 and 2 and helps us in dealing with noisy data in Table 5. This is only a hypothesis, and we may have a better answer in future work.
>
> 5) Regarding max-margin training.
> Thank you very much for the suggestion. Do you mean this paper http://www.jmlr.org/papers/volume10/xu09b/xu09b.pdf? Please advise. We will study the connection for future work, and we also want to see if it is appropriate to cite in this paper.
>
> Please let us know if we have missed anything and we'd be happy to continue the discussion.

---

> > ### Comment · AnonReviewer2 · 2018-11-30
> > **Yes, Xu et al. (2009) from JMLR**
> >
> > Yes, the reference http://www.jmlr.org/papers/volume10/xu09b/xu09b.pdf is the one I was referring to.
> >
> > Thank you for your answers. That addresses most of my concerns, though I may have to think more carefully about the loss function.

---

> > > ### Author Response · Authors · 2018-11-30
> > > **thank you very much**
> > >
> > > We would like to thank the reviewer for the upgraded rating and for the kind comments. And we thank the reviewer again for the valuable suggestions that help us improve this paper and future work. We'd be happy to answer further questions on the loss function.

---

> ### Author Response · Authors · 2018-11-10
> **our responses, part 1 of 2**
>
> We thank the reviewer for the thoughtful review and helpful suggestions. We are updating the paper to incorporate some of the suggestions and will post a revision soon. In the responses below, related points are grouped together and ordered roughly by significance.
>
> Before going into the list, we wish to emphasize that this paper sets a new state of the art in adversarial defense. For security and for generalization, robustness in terms of L2 norm and L_inf norm are both important. As the reviewer pointed out, notable defense progresses in the field so far have been mostly against L_inf-bounded attacks, except for some results in Madry et al. (2017). L2 defense is a less understood, and perhaps more difficult, problem than L_inf. Since both attack types are equally valid and there has been less advances on L2 defense, that makes any work in that area more important, and our paper represents a big step forward.
>
> 1) Regarding the loss function.
> The reason that (4) can express cross-entropy loss of an ordinary network is the following. Given any ordinary ReLU network without weight regularization, pick one layer, if we divide the weight matrix of this layer by a constant c, and divide the bias vectors of this layer and all subsequent layers by the same c, and we multiply the final logits by the same c, then there would no change in the end-to-end behavior of this network. The only change from the above is that the internal activations from that layer on are all scaled by 1/c. If we do the above for all layers and choose c=sqrt(b(W)) for each layer, where b(W) is from equation(2), we can convert the initial ordinary network to a nonexpansive network, only now with extra multipliers on the logits. After considering split layers (even when there is no split layers, we treat the last linear layer as a split layer, see first paragraph of Section 2.4), the multipliers on each logit become different. Therefore, the cross-entropy loss of the initial ordinary network is equal to term (4) of the nonexpansive network with proper u_1,...u_K values.
> The average in equation (6) is averaging over a batch.
>
> 2) Regarding L2 robustness and L_inf robustness.
> As the reviewer kindly pointed out, we report defense results against both L2-bounded attacks and L_inf-bounded attacks. For L2, L2NNNs set a new state of the art in Tables 1 and 2. At the same time, L2NNNs exhibit, in Table 4, near-state-of-the-art defense against L_inf-bounded attacks.
> It is true that Model 2's, from Madry et al. (2017), were trained with an L_inf adversary. However, let us quote from Madry et al. (2017): "our MNIST model retains significant resistance to L2-norm-bounded perturbations too -- it has quite good accuracy in this regime even for epsilon=4.5." and "our networks are very robust, achieving high accuracy for a wide range of powerful adversaries ..." In other words, Madry et al. do not see the use of L_inf attacker in training as a limiting factor to L2 defense.
> We are not aware of any published defense results that beat Madry et al. (2017) as measured by any norm. Please see also Athalye et al. (2018) for a comparison between Madry et al. (2017) and a set of other defense works. We are also not aware of any published MNIST or CIFAR models that were trained with an L2 adversary and achieved sizable white-box defense, and we ourselves have not found an efficient way to train with an L2 adversary.
> Another fact to consider is that our Model 4's were trained with the same L_inf attacker and that improved L2 robustness as reported in Tables 1 and 2.
>
> 3) Regarding confidence gap and robustness bound.
> The reviewer is correct that we could have chosen to report provable robustness rather than measured robustness, by using the noise bound guarantee provided by confidence gaps. If we had chosen a smaller L2 epsilon, say 2 for MNIST, our Model 3 has a provable robustness of 17.0%; if we chose 1.5, our Model 3 has a provable robustness of 46.5%.
> The reason that we chose measured robustness can be seen on examples in Figure 3. For each of the images, the guaranteed bound on noise L2-norm is half of the gap value, yet in reality the noise magnitude needed is much larger, 1.5X to 2X larger, than the guarantee. The reason is that the true noise bound is a function of local Lipschitz constants, as pointed out by Hein & Andriushchenko (2017), and local Lipschitz constants can be substantially below 1 in our models. The true bound is prohibitive to compute except for very small networks.
> Therefore, in order to demonstrate our defense with a more meaningful L2 epsilon of 3 and also compete with Madry et al. (2017) on the larger epsilon, we chose to report measured robustness.

---

### Public Comment · (anonymous) · 2018-10-02
**Clarification question**

Section 3.1 compares to the classifier from Madry et al 2017 as state of the art. The classifier from Madry et al was trained to resist L_infty perturbations but this work studies L_2 perturbations. Is your Model2 in Table 1 a checkpoint downloaded from Madry et al's website (trained to resist L_infty) or a version of their model that has been retrained to resist L_2? Sorry if this is already explained in the paper somewhere.

---

> ### Author Response · Authors · 2018-10-02
> **our answer to the question**
>
> Thanks for the comment and we're happy to clarify.
>
> Model 2's in Tables 1 and 2 were downloaded from Madry et al.'s GitHub pages (links in footnote on page 4) and were fetched under name "secret": these were released after they closed the black-box leaderboards and match what were reported in their paper.
>
> It is true that Model 2's were trained with L_inf adversary. However, let us quote from Madry et al. (2017): "our MNIST model retains significant resistance to L2-norm-bounded perturbations too -- it has quite good accuracy in this regime even for epsilon=4.5." and "our networks are very robust, achieving high accuracy for a wide range of powerful adversaries ..." In other words, Madry et al. do not see the use of L_inf attacker in training as a limiting factor to L2 defense.
>
> We are not aware of any published defense results that beat Madry et al. (2017) as measured by any norm. Please see also Athalye et al. (2018) for a comparison between Madry et al. (2017) and a set of other defense works. We are also not aware of any published MNIST or CIFAR models that were trained with L2 adversary and achieved sizable white-box defense.
>
> Another fact to consider is that our Model 4's were trained with the same L_inf attacker and that improved L2 robustness as reported in Tables 1 and 2.

---

### Public Comment · (anonymous) · 2018-10-02
**Clarification question**

Section 2.4 says "For a classifier with K labels, we recommend building it as K overlapping L2NNNs, each of which outputs a single logit for one label. In an architecture with no split layers, this simply implies that these K L2NNNs share all but the last linear layer and that the last linear layer is decomposed into K single-output linear filters, one in each L2NNN". How is this different from a normal neural network that uses a matrix with K columns for the final output layer?

---

> ### Author Response · Authors · 2018-10-02
> **our answer to the question**
>
> Thanks for the comment and we're happy to clarify.
>
> The difference is not in architecture, but rather in how to regularize the last linear layer: we recommend treating the last layer as K independent filters rather than regularizing it as a single matrix. In a single-L2NNN classifier, the K logits become related to each other: for example, if one activation in the second-last layer increases by 1, with all other activations staying the same, the K logits as a vector can only change up to L2 norm of 1. In contrast, in a multi-L2NNN approach, each individual logit can increase or decrease up to 1.
>
> We empirically observe that the multi-L2NNN approach results in better robustness than viewing the whole classifier as a single L2NNN.
>
> We also empirically observe that having split layers, i.e., final separate stacks of layers where each stack computes a single logit, helps improve the performance. In the multi-L2NNN approach, each such stack is covered by one of the K L2NNNs.

---

### Public Comment · (anonymous) · 2018-10-02
**Clarification question**

Does this paper contain an evaluation of the model in the max norm threat model? I haven't been able to find one. My guess is that this model does not work particularly well on standard benchmarks. Madry et al 2017 evaluate using a max norm ball of size .3. The largest L2 perturbation that fits within this max norm ball has size sqrt(.3^2*784) = 8.4. The evaluations in this paper use L2 norm with size 1.5. My guess is that the method proposed here can't scale to the size used by previous work.

---

> ### Author Response · Authors · 2018-10-02
> **our answer to the question**
>
> Please see Table 4 on page 6. For L_inf defense, we are on par with Madry et al. (2017) for CIFAR-10, and Madry et al. (2017) is better on MNIST.
>
> It seems that the commenter may have misread Madry et al. (2017). They use L_inf epsilon of 0.3 for MNIST and 8/256 for CIFAR-10, not 0.3 for both.
>
> We disagree with how the commenter translates L_inf epsilon to L2 epsilon. As show in Table 1, Model 2 can barely defend against L2 epsilon of 3, not to mention 8.4. For individual examples, please see Figures 1 and 5.

---

### Public Comment · (anonymous) · 2018-10-02
**Thought experiment: classifying MNIST**

It seems to me that it's not possible to achieve robustness against adversarial examples using Lipschitz smoothness, without also losing the ability to classify clean data. This claim is problem-dependent, but applies to the MNIST dataset used in this paper.

See https://arxiv.org/pdf/1806.04169.pdf Fig 18.
This figure shows that there is an MNIST 4 and an MNIST 9 that are only L2 distance 2.83 apart. The figure also shows a clean 4 and a mildly noisy 4 separated by small random noise with L2 norm 4.8. If we want the classifier to be so L2-smooth that it is guaranteed to assign the same class to the clean 4 and slightly noisy 4, then it must also assign the same class to the 4 and the 9. In other words, we'd like the function to be smooth in the vicinity of the 4 and smooth in the vicinity of the 9, but somewhere between the 4 and the 9 it needs to be significantly less smooth.

Do you know of a way around this problem?

---

> ### Author Response · Authors · 2018-10-03
> **our answer to the question**
>
> Thanks for the comment and we're happy to clarify.
>
> The commenter's thought experiment is correct. However, the conclusion is not about Lipschitz smoothness, but rather about all classifiers, including humans. Indeed it is impossible for a human to have 100% accuracy on clean MNIST images and at the same time 100% after-attack accuracy with noise L2 norm limit of more than 2.83/2 = 1.42. Because there exists a point in the input space that is 1.42 distance away from an image of 4 and also 1.42 distance away from an image of 9.
>
> Therefore, we would like to rephrase the commenter's question to the following. What is a reasonable goal for a robust classifier? The answer, in our opinion, is one that mimics a reasonable human. A human can have at or near 100% accuracy on clean MNIST images, but would have a different degree of confidence for each individual image. For the said two images of 4 and 9, his/her confidence should be low, -- using our terminology in the paper, confidence gap would be less than 1.42*sqrt(2), -- while for the vast majority of MNIST images, his/her confidence should be high. Consequently his/her after-attack accuracy is less than 100%, which is perfectly fine. It's not the fault of the classifier but the fault of certain ambiguously written digits.
>
> In this paper, we evaluate MNIST classifiers with noise L2 norm limit of 3. This implies that an attacker can modify nine pixels from pure white to pure black or vice versa, and it can modify more pixels with less swings. We believe that a human would have enough confidence to defend against this noise magnitude for the vast majority of MNIST images. If we were to speculate, a human would have after-attack accuracy of over 95%. That's the goal in our opinion and not 100% after-attack accuracy.
>
> Before our work, the state of the art is 7.6% or less, as shown in Table 1. We advance that to 24.4%. Although this is still a far cry from 95%, it is a big step up from 7.6%, and is better than all existing techniques as far as we know.
>
> By the way, the commenter's thought experiment is much related to why L2NNNs generalize well from noisy data with partially random labels. Please see Section 3.3, and in particular Table 6.
>
> Please also see the second paragraph on page 2 about preserving distance. The noise with norm 4.8 that the commenter mentioned is an example of a distance that is likely lost, while the distance of 2.83 mentioned is an example of a distance that we want to preserve through an L2NNN.

---

### Public Comment · ~Robin_Tibor_Schirrmeister1 · 2018-10-12
**Two-sided RELU = Concatenated RELU**

I wanted to make the authors aware that the proposal in section 2.2:
"We propose two-sided ReLU which is a function from R to R^2 and simply computes ReLU(x) and ReLU(-x)."

Had been proposed before as Concatenated ReLUs:  "Understanding and Improving Convolutional Neural Networks via Concatenated Rectified Linear Units" https://arxiv.org/abs/1603.05201
It seems people also try this scheme on ELUs already:
https://github.com/openai/weightnorm/blob/dff0cd132e9c6e0a31b76cb243d47a07e0c453cc/tensorflow/nn.py#L12-L15

---

> ### Author Response · Authors · 2018-10-12
> **thank you and we will add the reference**
>
> Hi Robin,
>
> Thank you very much for the reference and we will cite it in Section 2.2.
>
> It is interesting that what we call two-sided ReLU has shown values outside of the scope of adversarial robustness. Perhaps between our paper and the one you pointed out, it will become more accepted. There are a couple of differences which add to the synergy. We use two-sided ReLU for a different purpose (preserving distance for better robustness) and hence do not limit it to just convolution layers. We also propose a generalized scheme in Section 2.2 which can convert nonlinearities other than ReLU to two-sided forms which are nonexpansive and preserve distance better than the original nonlinearities.

---

### Public Comment · (anonymous) · 2018-10-29
**Very well done evaluation**

This paper has a thorough and well done evaluation section. I have one question about your evaluation: while it's not directly related to your paper, I wonder if you have insights about why the model from Madry et al. (2018) appears to continuously get worse as the number of iterations increases (even up to a million iterations!). I would have expected that the number of iterations wouldn't need to get this large. Do you think the model is somehow making gradient descent harder?

(Also: I would like to thank the authors for releasing their pre-trained models. I was able to download and evaluate these models, and so far haven't been able to reduce the robustness claims made in the paper.)

---

> ### Author Response · Authors · 2018-10-30
> **our answer to the question**
>
> Thank you very much for the comments and for trying out our models.
>
> It is an excellent question and one we've wondered about. Our speculation is indeed that adversarial training makes gradient descent harder, as it has an effect of flattening gradients around training data points. Madry et al. (2017) is the best we are aware of, and for the MNIST classifier, their adversarial training was so successful that we suspect that gradients are near zero in some parts of the input space, and hence it takes more iterations for an attacker to make progress. In other words, we suspect that, within many linear sections of their ReLU network, the logits have nearly flat values. The results suggest that adversarial training alone does not achieve full coverage around original-image points, and linear sections with large gradients still exist and hence bad points do exist nearby. It becomes a question of after how many steps does an attacker guided by gradient descent stumble close enough to one bad point. By the trend in Table 1, it would not be surprising if 10 million iterations would knock the accuracy down further.
>
> Actually we are intrigued and will do some gradient measurements which might put some numbers behind the speculation.

---

> > ### Author Response · Authors · 2018-11-01
> > **more**
> >
> > It turns out that comparing logit-to-image gradients across classifiers is harder than we thought. The issue is that logits need to be scaled properly to have a meaningful comparison of gradient magnitudes, and there does not seem to be a rigorous way to do so. However, we do think that the hypothesis stated above is plausible.

---

### Public Comment · (anonymous) · 2018-11-09
**related work**

How does your work relate to the following papers?

Miyato, et al. Spectral Normalization for Generative Adversarial Networks.   ICLR’18.

Tsuzuku, et al. Lipschitz-Margin Training: Scalable Certification of Perturbation Invariance for Deep Neural Networks. NIPS’18.

K. Scaman and A. Virmaux. Lipschitz regularity of deep neural networks: analysis and efficient estimation. NIPS’18.

Gouk, et al.  Regularisation of Neural Networks by Enforcing Lipschitz Continuity. Arxiv, 1804.04368.

Sedghi, et al.  The singular values of convolutional layers.  Arxiv, 1805.10408.

---

> ### Public Comment · (anonymous) · 2018-11-09
> **Two of those papers seem unrelated**
>
> I don't see how these two papers are related at all: the former is about GANs, and the second is talking about increasing accuracy, and doesn't mention robustness at all.
>
> Miyato, et al. Spectral Normalization for Generative Adversarial Networks.   ICLR’18.
>
> Sedghi, et al.  The singular values of convolutional layers.  Arxiv, 1805.10408.

---

> ### Author Response · Authors · 2018-11-12
> **our answer to the question**
>
> We are happy to comment on relations to these works.
>
> The Miyato et al. paper has a different way of approximating the spectral radius of a weight matrix. In place of the strict bound of (2) in our paper, they approximates the current spectral radius based on a companion vector, which is intended to approximate the top singular vector at the moment and which is updated through power iterations. The up side of their approach is that it can be computationally cheap, in fact they do just one power iteration to the companion vector after each training batch. The down side of their approach is that their spectral radius is a coarse approximation: for example, consider a scenario where two top singular values are close in magnitude, and the companion vector represents v1, one of the two corresponding singular vectors v1 and v2; their regularization would suppress the first singular value, and after a few batches the second singular value becomes dominant; at this point, it would take many power iterations to move the companion vector from v1 to v2, and one power iteration per batch certainly would not make it. When there are more singular values with similar magnitudes, the situation gets even worse. The end result is likely that their models are expansive. The empirical results suggest that they improve GANs, it's unclear how they would perform under adversarial attacks.
>
> The Tsuzuku et al. paper differs from us in a number of ways. To estimate Lipschitz constants, they follow Miyato et al. and use the same companion-vector approach. As we explained earlier, this approach has its limitations. They have a different way of modifying the loss function than ours. There does not seem to be anything that competes with our architecture changes. Their MNIST results do not seem strong; we cannot comment further on their empirical results as they only reported CW attacks with 100 iterations.
> We would like to bring it to AnonReviewer1's attention that this is an adversarial defense work that gets accepted into NIPS with weaker results than ours.
>
> The Scaman and Virmaux paper is on analysis of Lipschitz constants and not on optimization. In terms of analysis, the big difference between their AutoLip and ours is that they calculate Lipschitz constants of linear layers through power methods rather than we using bound of (2). Note that there is no companion vector here as it's one-shot analysis. Power methods would give a tighter estimation than our bound of (2), however they are too expensive to use in training a neural network and hence do not have practical implications for building robust models. Their SeqLip algorithm gives tighter bound, but is even more expensive, than AutoLip, and similarly would be difficult to use in training models.
>
> The Gouk et al. paper also uses power method to estimate L2 Lipschitz constants of linear layers, and this is the main different from us in terms of regularization. We have not found an explicit statement on whether they use a companion vector or start from a random vector each time. Some statements suggest that they do use a companion vector like Miyato et al., for example, they mentioned for one experiment they only do one power iteration. As discussed earlier for Miyato et al., this approach has its limitations. In fact, the authors acknowledged multiple times in the text that they are underestimating the L2 Lipschitz constants of linear layers. There are no robustness results.
>
> The Sedghi et al. paper is an interesting paper and they invented a way to compute the Lipschitz constant of a convolution layer. If their proof is correct, it would produce a tighter bound than ours, with a complexity that is lower than power methods. It seems that the computation cost is still fairly high, and hence they use it once every 100 iterations to regularize the convolution layers, and that resulted in improved nominal accuracy on CIFAR-10. It is unclear whether this is applicable in training robust models, especially if we can only afford to do it once in a while, -- it might be and is worth looking into and we thank the commenter for the reference. There are no robustness results in their paper. Note also that this is for convolution layers only.

---

### Public Comment · ~Aurko_Roy1 · 2018-11-15
**Question on PGD training from Madry et al**

When computing the PGD step for the model from Madry et al, do you use sign(gradient) or do you normalize the gradient by it's \ell_2 norm, when taking a single step? If you use the former then it seems like an unfair comparison since you are attacking it in an \ell_2 ball. In any case it would be interesting to see how PGD training w.r.t to the \ell_2 ball (so that you normalize the gradients by the \ell_2 norm) compares to the proposed method.

---

> ### Author Response · Authors · 2018-11-15
> **our answer to the question**
>
> Hi Aurko,
>
> Thank you for the interest.
>
> As we stated in the paper, Model 2's in Tables 1 and 2 were downloaded from Madry et al.'s GitHub pages (links in footnote on page 4). To be more specific, they were fetched under name "secret": these were released after they closed the black-box leaderboards and match what were reported in their paper.
> It is true that Model 2's were trained with L_inf adversary. However, let us quote from Madry et al. (2017): "our MNIST model retains significant resistance to L2-norm-bounded perturbations too -- it has quite good accuracy in this regime even for epsilon=4.5." and "our networks are very robust, achieving high accuracy for a wide range of powerful adversaries ..." In other words, Madry et al. do not see the use of L_inf attacker in training as a limiting factor to L2 defense.
> We are not aware of any published defense results that beat Madry et al. (2017) as measured by any norm. Please see also Athalye et al. (2018) for a comparison between Madry et al. (2017) and a set of other defense works. We are also not aware of any published MNIST or CIFAR models that were trained with L2 adversary and achieved sizable white-box defense.
> Another fact to consider is that our Model 4's were trained with the same L_inf attacker (PGD with default hyperparameters from Madry et al.'s GitHub) and that improved L2 robustness as reported in Tables 1 and 2.
>
> It is unclear that your suggestion would work in practice. The first question to ask is should one clip after all PGD iterations or clip per iteration.
> If one chooses to clip after all PGD iterations, then this new adversary is not much different from PGD with a smaller L_inf epsilon, and it's more likely to weaken the effect of adversarial training than help it.
> If one chooses to clip per iteration, then for each PGD iteration, we need to solve for the crossing point between a sphere and a line, where the line does not cross the center of sphere except for the very first iteration, and where the sphere has been modified by value range of each input entry. This is a quadratically constrained quadratic programming problem, and solving it per iteration would make PGD adversarial training much more expensive if not prohibitive, and it is difficult to implement on GPU.
> But by all means, we'd encourage you to do so, improve on Madry et al. (2017)'s L2 defense, and publish if it succeeds.
>
> Again, we are not aware of any published models from successful adversarial training with L2 adversary. We ourselves have made an unsuccessful attempt to use CW L2 attack in training, and it did not work because L2 attacks with low iteration counts do not seem to help our models yet we cannot afford L2 attacks with high iteration counts in the training loop. As a result, we decided to use the original PGD to build our Model 4's in Tables 1 and 2, and that gave them a nice boost in L2 robustness.
>
> We would be very interested if someone demonstrates successful adversarial training with L2 adversary, as we want to learn from him/her to improve our Model 4's and we would be happy to include more competitors in Tables 1 and 2.

---

> > ### Public Comment · ~Aurko_Roy1 · 2018-11-16
> > **L2 adversarial training has been done before**
> >
> > See https://arxiv.org/pdf/1805.12152.pdf (Figure 1)

---

> > > ### Author Response · Authors · 2018-11-16
> > > **that data support our argument**
> > >
> > > Hi Aurko,
> > >
> > > Thank you and that is precisely the kind of runs we were looking for. A closer look at their numbers actually reinforces our argument.
> > >
> > > According to Table 4 in https://arxiv.org/pdf/1805.12152.pdf, the best MNIST L2 robustness by training with L2 adversary is 63.73% robust accuracy against epsilon of 2.5.
> > > According to Figure 6 in Madry et al. (2017), the L2 robustness by training with L_inf adversary is about 90% against epsilon of 4.
> > > Note that both papers are from the same authors.
> > > In other words, by their own assessment, training with L_inf adversary produces stronger L2 defense than training with L2 adversary.
> > > This exactly supports our argument and is consistent with our own experience as stated in our last response.
> > >
> > > Having said the above, there is clearly something that Madry et al. know while we do not, which is how to convert PGD to an efficient L2 adversary, and we will try and find out.

---

> > > > ### Public Comment · ~Aurko_Roy1 · 2018-11-16
> > > > **I am not convinced l_\infinity training is better for l_2 adversary**
> > > >
> > > > If you look at Table 4 of Tsipras et al, on MNIST with an epsilon of 2.5 an l2 trained model gets 63.73% robustness (as you point out), while the l_\infty trained model on epsilon 3.0 from Madry et al gets 7.6% while L2NN gets 24.4% (Table 1 of this submission). Similarly on CIFAR-10 with an epsilon of 1.25 an l2 trained model of Tsipras et al gets 39.76% (Table 4), while from your Table 1 the l_\infty trained model with epsilon 1.5 from Madry et al gets around 9% while L2NN gets 20.4%.
> > > >
> > > > Granted the epsilon of Tsipras et al is slightly smaller than in your case, still the robustness of the l_\infty trained model is orders of magnitudes smaller than the l_2 trained model against an l_2 adversary. The robustness of L2NN also seems much worse (epsilon is slightly more, so it is not strictly comparable), but that's why I believe this work is missing a crucial baseline - comparison to an l_2 adversarially trained model.

---

> > > > > ### Author Response · Authors · 2018-11-16
> > > > > **equal-attack is the basis of comparison**
> > > > >
> > > > > Hi Aurko,
> > > > >
> > > > > When comparing two models, one has to decide on a common setup.
> > > > >
> > > > > If one chooses Madry et al.'s setup of L2 defense evaluation, then the comparison is as we stated:
> > > > > Training with L_inf adversary produces 90% against epsilon of 4.
> > > > > Training with L2 adversary produces 63.73% against epsilon of 2.5.
> > > > >
> > > > > If one chooses our setup of CW attack with high iteration count, then both the above numbers will reduce.
> > > > > Training with L_inf adversary produces 7.6% against epsilon of 3.
> > > > > Training with L2 adversary produces ?.
> > > > >
> > > > > Unfortunately we do not have access to their L2-adversary-trained model to fill in the question mark above. If one believes that Madry et al.'s setup of L2 defense evaluation extrapolates, that question mark is likely a very small number.

---

### Public Comment · (anonymous) · 2018-11-15
**Estimates of the Lipschitz constant**

I enjoyed reading this paper. I have a couple questions.

First, to reiterate on the comment below, during adversarial training, do you adversarially perturb in the signed gradient direction (FGSM), or the L2-normalized gradient direction? It seems to me because you are measuring robustness in L2, the latter should be used. I doubt FGSM will be effective, since this is not necessarily perturbing images in a large L2 measurable direction.

Second, you hypothesize your method reduces the local Lipschitz constant (modulus of continuity). Do you have any hard evidence to support this claim? There have been several papers now which provide reasonable and effective ways of estimating the Lipschitz constant of a network, at least locally. One simple method is to calculate the L2 norm of the Jacobian on a subset of the test images. The confidence gap is only part of the picture, but it does not correspond directly to the Lipschitz constant. I'd hope that you should see a noticable decrease in the L2 norm of the Jacobian using your regularization method.

---

> ### Author Response · Authors · 2018-11-15
> **our answer to the question**
>
> Thank you for the interest and we're happy to clarify.
>
> For the first point, please see our answer to Aurko Roy's comment.
>
> For the second point, we want to clarify a few things. We have a guarantee that the Lipschitz constant of an L2NNN is strictly no great than 1. Our hypothesis on local Lipschitz constant is regarding the effect of adversarial training on L2NNNs. Initially we expected that the gap between Model 3 (L2NNN with no adversarial training) and Model 4 (L2NNN with adversarial training) will fade away as CW attacker uses more iterations. Tables 1 and 2 suggest the opposite, i.e., that the benefit of adversarial training is permanent on L2NNN and is not just making examples difficult to find. To explain this phenomenon, we cite Hein & Andriushchenko (2017) and hypothesize that adversarial training on L2NNNs reduces local Lipschitz constants and thereby enlarges the actual robustness ball.
>
> We will measure L2 norm of Jacobians, albeit only as a surrogate for local Lipschitz constants, and report back here. Please give us a day or two, we want to finish a revision first.

---

> > ### Public Comment · (anonymous) · 2018-11-16
> > **thanks for reply and follow up question**
> >
> > Great, thanks for your speedy response. That clarifies my question. I'm very curious to see evidence that the local Lipschitz constant is reduced via your method.
> >
> > One follow up question -- are your l2 distances measured on [0,255] pixels, or pixels normalized to [0,1]?

---

> > > ### Author Response · Authors · 2018-11-16
> > > **measurements**
> > >
> > > Let us answer your later question first. Pixels are normalized to [0,1] for all runs in our paper, for both MNIST and CIFAR.
> > >
> > > Now, as promised, the following are average L2 norm of Jacobians of logits with respect to inputs, averaged over the first 1000 images in MNIST test set.
> > >
> > > Model 2 (Madry et al. (2017)): 10.818453
> > > Model 3 (L2NNN with no adversarial training): 1.054181
> > > Model 4 (L2NNN with adversarial training): 0.8331261
> > >
> > > A few things to note:
> > > -- This is a surrogate for local Lipschitz constants, in particular it is measured at the nominal point and not over a neighborhood.
> > > -- L2 norm of Jacobian for Models 3 and 4 can be larger than 1, because we built them as multi-L2NNN classifiers, please see the first paragraph of Section 2.4.
> > > -- The comparison between Model 3 and Model 4 is consistent with our hypothesis.
> > > -- We would not take the Model 2 number at face value, because one could argue that Model 2 should be scaled down by a constant before making this measurement. This scaling is a tricky issue, please see our response to an earlier comment titled "Very well done evaluation".

---

> > > > ### Public Comment · (anonymous) · 2018-11-19
> > > > **thanks for checking Jacobian norm**
> > > >
> > > > Great, thank you for checking those Jacobian norms. Reporting those numbers very much strengthens your results.

---

> > > ### Public Comment · (anonymous) · 2018-12-13
> > > **Normalized for CIFAR-10 also?**
> > >
> > > This doesn't make sense. If you look at Figure 6 in https://arxiv.org/pdf/1706.06083.pdf , at eps=100 the accuracy is nearly zero and note that this is at less than 100 steps of PGD. That eps=100 from Madry et al, corresponds to eps=100/255 if you normalize it (note that Madry's CIFAR pixels are not normalized).
> > >
> > > In your paper, at eps=1.5 (in Madry's scaling, it's eps=380+), you quote ~14% accuracy for Madry. Isn't this strange? That even after you increase the budget the nearly 4 times, your accuracy is still quite the same.

---

> > > > ### Author Response · Authors · 2018-12-13
> > > > **likely a mistake in Fig 6(d) in Madry et al. (2017)**
> > > >
> > > > We too were confused about Fig 6(d) in Madry et al. (2017), until we see
> > > > https://arxiv.org/pdf/1805.12152.pdf
> > > > which is from the same authors. There is evidence that they made mistakes in computing x coordinates when plotting Fig 6(d) in Madry et al. (2017), and underreported the robustness.
> > > >
> > > > The last two rows in Table 4 in
> > > > https://arxiv.org/pdf/1805.12152.pdf
> > > > show substantial robustness under l2 epsilon of 80/256 and 320/256. These results suggest that 100/256 would be a low bar, and the weak curve of Fig 6(d) in Madry et al. (2017) is highly unlikely.
> > > >
> > > > Now consider our measurements of Madry model in our Table 2. The numbers are in line with numbers in
> > > > https://arxiv.org/pdf/1805.12152.pdf
> > > >
> > > > So the most plausible explanation is that the authors used incorrect x coordinates when plotting Fig 6(d) in Madry et al. (2017). If we were to venture a guess, maybe they only added up deltas in the red channel rather than all three RBG channels.

---

> > > > > ### Public Comment · (anonymous) · 2018-12-13
> > > > > **Not convinced**
> > > > >
> > > > > Have you checked with the authors this is the case?
> > > > >
> > > > > Also, Table 4 has no entries for the l2-accuracy for an l-inf trained model I think. Correct me if I am wrong. So, there is no "comparable" version that corresponds to the numbers in the first version.
> > > > >
> > > > > Also, your Table indicates 91.7% for eps=1.5 at 100 iterations, while Table 4 suggests 40% at eps=1.25 (I suspect it would be much worse tat eps=1.5). While they don't provide code, I am not sure the number of iterations for the entries in Table 4 is much more than 100.

---

> > > > > > ### Author Response · Authors · 2018-12-14
> > > > > > **it seems that you read the wrong table**
> > > > > >
> > > > > > 91.7% is from our Table 1, and it is for Madry's MNIST model with l2 epsilon of 3 and after 100 CW iterations. The number you were looking for is 13.9% in our Table 2.
> > > > > >
> > > > > > It is true that Madry et al's two papers are not reporting on the same cifar models. However the contrast between below-10% robust under l2 epsilon of 100/256 and 39.76% robust under l2 epsilon of 320/256 is so large that one of the two is likely incorrect. Our measurement of their model suggests that Fig 6(d) in Madry et al. (2017) is likely the incorrect one.
> > > > > >
> > > > > > Please also see our discussion with Aurko Roy, particularly the last few rounds regarding training with l2 adversary vs training with linf adversary.

---

> > > > > > > ### Public Comment · (anonymous) · 2018-12-14
> > > > > > > **Continuing discussion**
> > > > > > >
> > > > > > > Oops, I did indeed read the wrong table. Thanks for pointing that out.
> > > > > > >
> > > > > > > To continue the discussion, your responses to Aurko Roy seem to be centered around MNIST. From what I believe, and what has been advocated in recent literature on adversarial defenses, MNIST should only be used as a "sanity check". Here, you seem to be making the claim that in general, across datasets, l2 training is worse than l-inf training. Can you confirm this for other data-sets?
> > > > > > >
> > > > > > > I'm curious because it seems highly unintuitive that l2 training would do worse than l-inf training for l-2 robustness.
> > > > > > >
> > > > > > > The eps at which adversaries are generated governs the robustness of the model. Have you tried varying the size of the l-inf/l-2 balls for Madry et al. style training and see if the numbers don't improve much? I strongly feel that is a hyperparameter that needs to be tuned, more so when comparing the performance of an l-inf defense against an l-2 attack.
> > > > > > >
> > > > > > > Would you be open to making your models open-source?

---

> > > > > > > > ### Author Response · Authors · 2018-12-14
> > > > > > > > **our models are at the Dropbox link on page 4**
> > > > > > > >
> > > > > > > > Please see the dropbox link at the first paragraph of section 3.
> > > > > > > >
> > > > > > > > One commenter has kindly tested our robustness numbers. Please see the comment titled "Very well done evaluation".
> > > > > > > >
> > > > > > > > In the paper we stated that our model 4's were trained with linf adversary, but we made no claim about training with linf adversary vs training with l2 adversary. This only came up in the discussion with Aurko as an empirical observation.
> > > > > > > >
> > > > > > > > We disagree with the statement that MNIST is only a sanity check. From robustness perspective, MNIST is far from a solved problem.

---

> > > > > > > > > ### Public Comment · (anonymous) · 2018-12-14
> > > > > > > > > **Agree with most your comments**
> > > > > > > > >
> > > > > > > > > Thanks for making things open-source! Greatly appreciate it.
> > > > > > > > >
> > > > > > > > > I agree with most things in your response. Yes, it came up because of the possibility that l2-training would make the network more robust to l2-perturbations than l-inf training.. which (l2-training) I feel should be used as a relevant baseline. To summarize, if I understand correctly, you did not use l-2 training as a baseline because you believe from past literature that l-inf training is better for l2-attacks. Is that a correct statement to make?
> > > > > > > > >
> > > > > > > > > I also raised another point: I think you might have missed this -- but have you tried studying the effect of varying the perturbation bound of the l-inf balls on the l-2 robustness in the Madry et al. setting? Since, you are not directly optimizing for l2-robustness, it makes sense to tune hyperparameters when making tangential comparisons.
> > > > > > > > >
> > > > > > > > > I do not want to criticize/attack your paper. I found it quite interesting and I am playing Devil's advocate to answer some questions I  have and tease out things for my better understanding. Thanks for engaging the discussion.
> > > > > > > > >
> > > > > > > > > PS. I would not be very inclined to believe an anonymous reader claiming to have invalidated or validated your robustness claims, especially. Because of the nature of the forum, I believe anonymous comments (including mine) should be taken with a grain of salt.

---

> > > > > > > > > > ### Author Response · Authors · 2018-12-15
> > > > > > > > > > **always happy to answer questions on our paper**
> > > > > > > > > >
> > > > > > > > > > Regarding the specific points:
> > > > > > > > > > -- We chose the best available baselines, i.e. classifiers from Madry et al. (2017), which happen to be trained with L_inf adversary. There seem to be no available models that are trained with L2 adversary, and the only paper that talked about such models, i.e. https://arxiv.org/pdf/1805.12152.pdf, reported L2 robustness that is weaker than L2 robustness achieved by training with L_inf adversary. Again more details are in the discussion with Aurko.
> > > > > > > > > > -- Our Model 4's were trained with L_inf PGD with default hyperparameters from Madry et al.'s GitHub. We did not tune the hyperparameters, and you are right that there is probably room for improvement.
> > > > > > > > > >
> > > > > > > > > > Of course one may question everything on this forum. Luckily numbers never lie, and that is why we publish our models so that anyone can verify our results.

---

### Author Response · Authors · 2018-11-16
**a revision is posted**

We would like to thank the three reviewers for the many helpful suggestions, and we are grateful for the extensive comments from others as well. We have made our best effort in revising the paper within the page limit for the main text and only enlarging the appendix. There are a number of places where we would have liked to elaborate more and we hope we will have a chance to do so using more space if this paper is accepted.

---

### Public Comment · ~Chris_Finlay1 · 2018-11-26
**Missing references**

I’d like to direct you to two papers which you may have overlooked.

The first is another submission to ICLR 2019, “Improved robustness to adversarial examples using Lipschitz regularization of the loss” ( https://openreview.net/forum?id=HkxAisC9FQ ). They have achieved better robustness results (in L2) than yours on CIFAR-10, by over 10% at L2 distance 1.5. Moreover their method doesn’t degrade test error on unperturbed images, whereas your regularized networks have over 20% test error on CIFAR-10. I’d hope that a properly robust network should not have significantly worse test error.  How do you distinguish your results from this other ICLR submission?

Second, there is an arXiv paper by Gouk et al, “Regularisation of Neural Networks by Enforcing Lipschitz Continuity” ( https://arxiv.org/abs/1804.04368 ) from this spring which has methods very similar to yours. They enforce either the L-1 or L-infinity norm of the weight matrices to be less than 1, which is nearly what you are doing (you enforce L-infinity on W W^t and W^t W). (You stated earlier in the comments that Gouk et al uses the power method, but from my reading of that paper this is not true – they use the explicit formulas for the L-1 and L-infinity norms.) And they enforce this constraint by a projection step, which if I understand your paper correctly, is very similar to what you are doing (your formula W’ = W / \sqrt(b(W)) ). Could you comment on the merits of your approach vis-a-vis Gouk et al?

In addition, Gouk et al accounts for batch normalization – do you? I can’t tell if you have taken batch normalization into account when you compute your matrix norms. In effect batch normalization post multiplies the weight matrices by a diagonal matrix, which needs to be factored in. If you haven’t, that may explain in part why your test errors are not very good on the unperturbed images.

A general comment about matrix norms, and your title. I can’t help but think that your title is a bit misleading: forcing the L-infinity weights to have norm less than one seems like a very crude way of also bounding the L-2 norm of the weight matrices. Although yes ||A||_2 <= ||A||_\infty, we also have that ||A||_\infty <= \sqrt n ||A||_2. So the gap in the first inequality between these two norms can be very large, especially when we’re talking about convolution matrices with many channels. It seems to me that such a harsh projection, when the gap between these two norms is large, would seriously degrade network performance in practice. This may also explain your  poor test errors.

---

> ### Author Response · Authors · 2018-11-27
> **our answer to the question**
>
> Hi Chris,
>
> There are a number of issues with
> https://openreview.net/forum?id=HkxAisC9FQ
>
> 1) Questionable evaluation: Inserting sigmoid layer before attack.
> The authors stated that "Prior to the final softmax layer, we found inserting a sigmoid activation function improved model robustness. In this case, the sigmoid layer comprised of first batch normalization (without learnable parameters), followed by the activation function t*tanh(x/t), where t is a single learnable parameter, common across all layer inputs."
> There is something wrong: Adding a final sigmoid layer should have zero effect on a network's robustness, because an adversarial example before would still be an adversarial example after.
> These statements suggest that the authors applied attacks on the sigmoid outputs, or even worse, that they may have applied attacks on the softmax outputs. Such evaluation setup is a form of gradient obfuscation, and it is well known to artificially slow down gradient-based attacks and create a false sense of security.
> The proper way of evaluating robustness, as measured by white-box defense, is to apply attacks on the logits themselves, i.e. the direct outputs of ReLU network (or any final computing layer). That's what we do.
>
> 2) Questionable evaluation: Attack setup.
> For evaluation, the authors use attacks implemented in Foolbox:
> https://arxiv.org/pdf/1707.04131.pdf
> https://github.com/bethgelab/foolbox
> which contains a modified version of CW attack. The authors stated that "Hyperparameters were set to Foolbox defaults." The Foolbox version of CW attack uses a default of 1000 iterations. Among all Foolbox attacks, the authors concluded that L2 PGD is the strongest, stronger than Foolbox CW, and therefore used L2 PGD in all reportings.
> First, the authors should have used the original CW code at https://github.com/carlini/nn_robust_attacks.
> Second, 1000 iterations are not enough to evaluate robustness, in light of the L2 robustness numbers from Madry et al. (2017) and Tables 1 and 2 in our paper. Neither PGD with low iteration count nor CW with low iteration count reveals the true robustness of a model.
> Point 1) above makes the situation even worse. With gradient obfuscation, 1000 CW iterations may become equivalent to 100 iterations or less.
>
> Considering 1) and 2), one has to take their numbers with a grain of salt.
>
> 3) Reliance on training data coverage.
> In our Section 4, we review related works as two big groups. The first group fortify a network around training data points: this includes both adversarial training like Madry et al. (2017) and gradient regularization like Ross & Doshi-Velez (2017). The second group bounds a network's responses to input perturbations over the entire input space. Our work belongs to the second group.
> Their proposed method is fairly similar to Ross & Doshi-Velez (2017), and belongs to the first group. The common weakness of the first group is the reliance on training data coverage. While works in this group are able to fortify parts of the input space, specifically flattening gradients around training data points, there exists little control over parts not covered by training data.
>
> Your statement of "...by over 10% at L2 distance 1.5" is cherry-picking data: even if accepting their numbers as they are (big question mark by themselves), the difference would be only 1.2%.
>
> Regarding the Gouk et al. paper, please see our response dated 11/12 to an earlier comment titled "related work".
>
> Regarding your comments on matrix norm. It seems that you were confusing WTW with W, please see the text around equation (2) including footnotes. Measurements show that L2NNN's 2-Lipschitz constant is not far below 1, please see our response dated 11/16 to an earlier comment titled "Estimates of the Lipschitz constant" and our Figures 3 and 4.

---

> > ### Public Comment · ~Chris_Finlay1 · 2018-11-28
> > **reply**
> >
> > Hi,
> >
> > Thanks for the reply.
> >
> > Regarding point 1, on that paper's final sigmoid layer. You say that adding a final sigmoid layer will have no effect, but I disagree. Adversarial examples exploit network instabilities by finding directions which push incorrect logits to be large. A final sigmoid layer could clip such growth before the incorrect logit dominates the other labels. That is not gradient obfuscation, it is a reasonable robustness measure. It seems that this is what the other paper is reporting: with the same attack and attack hyper parameters (how else can you fairly compare models?), models with a final sigmoid layer perform better.
> >
> > Regarding your point 2. I agree it is hard to make a direct comparison between your numbers and theirs. Whether or not CW is better than L2 PGD is debatable. I just checked the Foolbox specs. Foolbox uses only 10 iterations in its L2 attack, whereas Foolbox's default CW uses 1000. So, given that the other paper finds consistently that L2 PGD beats CW (with only 10 iterations vs 1000!) it's hard not to conclude that L2 PGD is a better attack. I can't comment on how Foolbox has tweaked their implementation of CW, but I'd hope that they are nearly equivalent.
> >
> > I don't think it is cherry-picking to say that the other model outperforms yours by 10% for models trained without adversarial training. That is what your and their tables say.
> >
> > Regarding 3. It is fair to say that your method accounts for the entire input space, whereas theirs accounts for only data sampled from the data distribution. I believe though that the other paper reports all its numbers on test data, not training data, with good results. Given that, it may be that stabilizing a network on only the training data empirically stabilizes on the test data, since these two datasets are supposed to be drawn from the same distribution. It could be that it is not necessary to stabilize a network on the whole input space, but only on the data manifold.
> >
> > I suggest you comment on the other ICLR submission's OpenReview page, so that those authors have a change to fairly rebut your criticisms.
> >
> > Regarding Gouk et al. The Gouk paper does not use power iteration, which you erroneously stated below! They penalize by the L-infinity norm, no power iteration needed. You are also penalizing by the L-infinity norm. They also divide (‘project’) each layer by this norm, same as you. You haven’t addressed how your paper differs from Gouk et al.
> >
> > I’m aware of the difference between ||W^t W||, and ||W||. Since ||W^t W||_\infty <= ||W||_1 ||W||_infinity, I don’t see why it is necessary to penalize by ||W^t W||_\infty when what you are really trying to control is ||W||_\infty. Why not just penalize by ||W||_\infty? What do you gain by penalizing with ||W^t W||? Is it that you want to control both the 1-norm and the max-norm? My main point here is: the gap between ||W||_\infty and ||W||_2 can be large in high dimensions. My feeling is that using the infinity-norm to control the 2-norm is problematic, and your poor test errors haven’t shaken this feeling.
> >
> > You have not addressed two of my main points. Regarding my point on batch norm. Do you account for batch norm when computing your matrix norms? And how do you justify your poor test error, on the natural images? Ideally a robust network will have good test error and good robustness properties.

---

> > > ### Author Response · Authors · 2018-11-28
> > > **more on gradient obfuscation**
> > >
> > > Hi Chris,
> > >
> > > Consider two classifiers: the first is f(x) and the second is sigmoid(f(x)).
> > > Suppose we have an adversarial example for the first classifier: f(x) classify x0 correctly but x0+delta incorrectly. In other words: argmax(f(x0+delta))!=argmax(f(x0)).
> > > Because sigmoid is monotonically increasing, we have argmax(sigmoid(f(x0+delta)))==argmax(f(x0+delta)) and argmax(sigmoid(f(x0)))==argmax(f(x0)). Therefore argmax(sigmoid(f(x0+delta)))!=argmax(sigmoid(f(x0))). In other words, x0+delta is also an adversarial example for the second classifier.
> > > That's why adding a final sigmoid layer should have zero effect on a network's robustness, and that's why their finding of "inserting a sigmoid activation function improved model robustness" is nothing but gradient obfuscation.
> > > According to Table 3 in their paper, their best model goes from 78.42% error to 100% error when tanh is removed.
> > >
> > > Your comments on L2 PGD vs CW describe exactly the phenomenon of gradient obfuscation. 10 PGD iterations being stronger than 1000 CW iterations suggests a situation of vanishing gradient. If CW is used properly 1000 iterations of it will be a much stronger attack than 10 PGD iterations.
> > > 10 iterations of any attack are far too few for a meaningful robustness evaluation.
> > >
> > > With gradient obfuscation and 10 PGD iterations, all measurements in https://openreview.net/forum?id=HkxAisC9FQ are meaningless.
> > >
> > > As we said in the last comments, the proper way of evaluation is to use CW before sigmoid or softmax and call it with high iteration counts. That is what we do.
> > >
> > > Your statement of "They have achieved better robustness results (in L2) than yours on CIFAR-10, by over 10% at L2 distance 1.5." is simply cherry picking data because it ignores our best model.
> > > It's a pointless comparison anyway given that their numbers come from gradient obfuscation and 10 PGD iterations.
> > >
> > > It seems that you have misread the Gouk et al. paper. Please see the last paragraph of Section 3.1 and first paragraph of Section 4.1 in their paper:
> > > https://arxiv.org/pdf/1804.04368.pdf
> > >
> > > It seems that you are still confusing WTW with W, and your statement of "what you are really trying to control is ||W||_\infty" makes no sense: our models are L2 nonexpansive, not L_inf nonexpansive. As we said in last comments, please see the text around equation (2) including footnotes. Measurements show that L2NNN's 2-Lipschitz constant is not far below 1, please see our response dated 11/16 to an earlier comment titled "Estimates of the Lipschitz constant" and our Figures 3 and 4.
> > >
> > > We addressed batch norm in the appendix, and one simply needs to divide the scaling factors by the max one.
> > > The trade-off between robustness and nominal accuracy has been discussed extensively in multiple discussions throughout this page, please read. This trade-off has been observed by previous works including adversarial training and adversarial polytope works, and it remains an open question whether such trade-off is a necessary part of life. However, one thing we know for sure is that gradient obfuscation is not the answer.

---

> > > > ### Public Comment · ~Chris_Finlay1 · 2018-11-29
> > > > **reply**
> > > >
> > > > Hi again,
> > > >
> > > > Regarding gradient obfuscation. In classification f(x) is a vector, not a scalar.  There is no ordering on vectors, and so there is no notion of monotonicity. So saying "sigmoid is monotonically increasing, we have argmax(sigmoid(f(x0+delta)))==argmax(f(x0+delta))" is false. Sigmoid is applied point wise to the *vector* f(x). So yes, you could use a point wise sigmoid layer to improve robustness. If f(x) was a scalar, I agree that adding a sigmoid would do nothing, but it isn't.
> > > >
> > > > It looks like the authors in the other ICLR paper also used black box attacks (the Boundary attack), and found the black box attack to be a weaker attack than using PGD. If adding a sigmoid truly was gradient obfuscation,  then the black box attacks should be able to get around the hypothesized obfuscation. But they don't. Again I suggest you comment on that paper so that the authors there can rebut your criticisms.
> > > >
> > > > Regarding Gouk et al: Straight from Section 4.1 you pointed to, they talk about using the L-1 and L-inf norms as well as the L-2 norm. In their experiments they compare all three norms.
> > > >
> > > > My point regarding L-2 and L-inf norms is this. To summarize your method: You want to ensure the 2-norm of W is less than 1. This equivalent to keeping the 2-norm of W^t W (and W W^t) less than one. Since the infinity norm bounds the 2-norm of a matrix, it suffices to keep the infinity norm of W^t W (or W W^t) small. Which leads to two questions:
> > > >
> > > > 1) Are you not concerned that the gap between the 2-norm and the infinity norm can be huge, growing in the worst case with the square root of the input dimension?
> > > >
> > > > 2)  Since all other matrix norms bound the spectral norm, why go to the work of computing W^t W and W W^t at all, when ||W||_2 is bounded by ||W||_infinity in the first place? Why not just control ||W|_infinity instead, as in Gouk et al?

---

> > > > > ### Author Response · Authors · 2018-11-29
> > > > > **our response**
> > > > >
> > > > > Hi Chris,
> > > > >
> > > > > We are sorry that we will stop responding to your questions. We feel that most readers of our paper and this page do not share your confusions, and further discussion would not help the purpose of this forum. We feel that our two rounds of comments are sufficiently clear.
> > > > >
> > > > > As you requested, we will put a pointer on https://openreview.net/forum?id=HkxAisC9FQ so that you or anybody else can continue the discussion there.

---

> ### Public Comment · (anonymous) · 2018-12-11
> **Unreasonable to ask comparisons to concurrent submissions**
>
> I think it is unreasonable and a poor direction for members of the community to start posting requests for comparisons to concurrent submissions, especially ones made to the same conference!
>
> To be clear I am in no way associated to this paper, but saw this while browsing.

---

### Meta-Review · Area_Chair1 · 2018-12-15
**practical ideas for ensuring robustness, albeit in a limited attack model**

**Confidence:** 4
**Recommendation:** Accept (Poster)

**Metareview:**


* Strengths

This paper studies adversarial robustness to perturbations that are bounded in the L2 norm. It is motivated by a theoretical sufficient condition (non-expansiveness) but rather than trying to formally verify robustness, it uses this condition as inspiration, modifying standard network architectures in several ways to encourage non-expansiveness while mostly preserving computational efficiency and accuracy. This “theory-inspired practically-focused” hybrid is a rare perspective in this area and could fruitfully inspire further improvements. Finally, the paper came under substantial scrutiny during the review period (there are 65 comments on the page) and the authors have convincingly answered a number of technical criticisms.

* Weaknesses

One reviewer and some commenters were concerned that the L2 norm is not a realistic norm to measure adversarial attacks in. There were also concerns that the empirical level of robustness of the network was too weak to be meaningful. In addition, while some parts of the experiments were thorough and some parts of the paper were well-presented, the quality was not uniform throughout. Finally, while the proposed changes improve adversarial robustness, they also decrease the accuracy of the network on clean examples (this is to be expected but may be an issue in practice).

* Discussion

There was substantial disagreement on whether to accept the paper. On the one hand, there has been limited progress on robustness to adversarial examples (even under simple norms such as the L2 norm) and most methods that do work are based on formal verification and therefore quite computationally expensive. On the other hand, simple norms such as the L2 norm are somewhat contrived and mainly chosen for convenience (although doing well in the L2 norm is a necessary condition for being robust to more general attacks). Moreover, the empirical results are currently too weak to confer meaningful robustness even under the L2 norm.

* Decision

While I agree with the reviewers and commenters who are skeptical of the L2 norm model (and would very much like to see approaches that consider more realistic threat models), I decided to accept the paper for two reasons: first, doing well in L2 is a necessary condition to doing well in more general models, and the ideas and approach here are simple enough that they might provide inspiration in these more general models as well. Additionally, this was one of the strongest adversarial defense papers at ICLR this year in terms of credibility of the claims (certainly the strongest in my pile) and contains several useful ideas as well as novel empirical findings (such as the increased success of attacks up to 1 million iterations).